# LLM-MATLOGIC: Executable Exchange Contracts for Knowledge-Graph Query Answering with Scoped Negation

Dezhuang Miao [1]   Xiaoming Zhang [1]   Bo Zhang [2]   Yibin Du [1]   Xiang Li [1]   Ruilin Zeng [1]   Yirui Qi [1]

## Abstract

LLM-to-KG systems frequently fail on exclusion-rich questions because natural-language negation is both *scope-sensitive* and *evidence-dependent*: it may constrain only one subgoal/branch and only certain supporting paths, yet such attachment is rarely explicit in text. We propose the *Executable Exchange Contract* (EEC) to bridge this gap, specifying scope-bound exclusions as executable control metadata exchanged between a specifier and an executor. Our executor, MATLOGIC, compiles exclusions into scope-local masks applied during multi-hop propagation and executes requests under a unified $\mathbf{P}{\to}\mathbf{N}{\to}\mathbf{C/D}$ schedule, ensuring exclusions are enforced before witness loss and branch entanglement. The system can also return compact witness pointers to keep support types distinguishable when needed. We evaluate on both structured complex queries and end-to-end natural-language KGQA, and introduce contract-aligned diagnostics that isolate errors from specification versus execution and verify the necessity of scoped enforcement.

## 1. Introduction

Exclusion is a first-class requirement in real queries over structured knowledge. People often ask for results that satisfy several conditions while ruling out specific cases (Angles & Gutierrez, 2016). For example, a film fan may ask for movies that feature an actor but not as a brief cameo. Importantly, such conditions "not" are rarely answered by simply removing an entity from the final list. The reason is that *the same answer can be supported in different ways*,

and the exclusion may target only certain types of support (Ren et al., 2023) (e.g., specific relations, paths, or witness attributes). Negation is easy to mishandle in LLM-based pipelines that translate natural-language requests into graph operations, often resorting to post-hoc filtering or overly broad enforcement that fails to respect evidence-dependent exclusions (D'Abramo et al., 2025). As LLMs increasingly serve as front-ends to knowledge graphs in search and decision workflows, getting such exclusions right becomes urgent (Luo et al., 2024; Wang et al., 2025). The key challenge is to turn everyday negation into constraints that can be executed at the right step of reasoning, rather than approximated by ad-hoc prompts or after-the-fact filtering (Choudhary & Reddy, 2023; Jeong et al., 2025).

Many current systems leave such exclusions implicit. In natural-language requests, modifiers like "not as a cameo," "without involving $X$," or "except when . . ." *rarely* specify precisely which subcondition they modify or which type of support they rule out (Pröllochs et al., 2020). As a result, the system must decide *where* the exclusion attaches in the computation (e.g., to a particular clause, a branch of reasoning, or a specific supporting relation). Equally important is *when* the exclusion is enforced. The *timing* of enforcing an exclusion determines whether a *scoped (local) negation* can be evaluated correctly (supporting witnesses must not be projected away too early), whereas the *merge policy* determines whether such a scoped negation is mistakenly *globalized*. Otherwise, a local exclusion may spill over and take a global effect. These issues are not about adding another logical operator, but about making exclusions scope-aware and executable. This motivates an explicit exchange interface that encodes exclusions as executable constraints.

To close this gap, we propose *Executable Exchange Contract* (EEC), which makes exclusion handling explicit *at the specifier-executor interface* as a structured executable request. EEC requires the *specifier* to state (i) *where* each exclusion attaches in the intermediate computation (its scope, e.g., a clause/branch), and (ii) *what* it excludes (forbidden entity sets or evidence conditions), instead of leaving these choices implicit in text. The *executor* then performs scope-aware logical operations with the structure as the boundary, which compiles exclusions into negation constraints and

[1]School of Cyber Science and Technology, Beihang University, Beijing, Beijing, China [2]School of Computer and Electronic Information / School of Artificial Intelligence, Nanjing Normal University, Nanjing, Jiangsu, China. Correspondence to: Xiaoming Zhang <yolixs@buaa.edu.cn>.

*Proceedings of the 43$^{rd}$ International Conference on Machine Learning*, Seoul, South Korea. PMLR 306, 2026. Copyright 2026 by the author(s).

enforces them during reasoning. The key challenge for the executor is to efficiently reason over subgraph structure while preserving scope correctness and enabling targeted exclusion; even a single subgraph may contain more than 200 triples. By separating what is specified from how it is executed, EEC stabilizes exclusion semantics across different specifiers and reduces reliance on iterative prompting to "get the negation right." We instantiate EEC with an LLM specifier and a matrix-based executor, MATLOGIC, which executes EEC requests under a unified schedule.

Our work is related to complex query answering (CQA) over KGs and natural-language KGQA, but differs in its interface-centric view of exclusions. On CQA side, query-embedding methods typically assume an *explicit* logical query as input, where negations are already part of the query and their scope is fixed by the query structure (Hamilton et al., 2018; Ren & Leskovec, 2020; Chen et al., 2022). On the KGQA side, recent systems synthesize executable programs (e.g. SPARQL) (Liu et al., 2024) or perform iterative retrieval-guided reasoning over the KG by decomposition of LLM-powered thought (Ma et al., 2024). Although effective, these pipelines often do not make exclusion attachment and timing explicit *as part of the exchange* between front-end specifier and the back-end executor, which is fragile for evidence-dependent negation. This lets us study negation handling as a contract-level property of exchange, rather than tying it to a particular query language or prompt.

As an intermediate contract between the specifier and the executor, EEC separates semantic and logical computation, which can enhance reasoning ability of specifier and enabling unified logical computation in a limited subgraph by the executor, known as the LLM-MATLOGIC framework. Concretely, MATLOGIC (executor) closes the loop between the EEC contract and execution by consuming only contract-relevant control metadata: the merge structure (how subgoals compose) and stable scope IDs (where exclusions attach). It turns the operator skeleton into a small execution plan, handling disjunction by union *within* groups and conjunction by intersection *across* groups, with each atomic subgoal tagged by a scope ID that determines where exclusions apply. On the KG backend, atomic subgoals are executed by sparse-matrix propagation over relation-constrained multi-hop neighborhoods. For scoped negation, MATLOGIC pre-compiles each banned set into a *scope-local reachability mask* and applies it during propagation, so that witnesses traversing forbidden entities are blocked *within the intended scope*, consistent with the canonical enforcement boundary. Finally, the executor composes intermediate evidence according to the plan, performs a final projection to obtain answer entity IDs, and can return compact witness pointers sufficient to recover one concrete surviving support when needed. Our contributions are:

- We formulate exclusion-rich negation in LLM-to-KG reasoning as an *exchange contract* between a specifier and an executor, separating *what* to exclude (with scope) from *how* it is executed.

- We identify a *canonical enforcement boundary* (**P→N→C/D**) for scoped exclusions by showing why moving negation across projection or disjunctive merging changes semantics for evidence-dependent negation.

- We instantiate EEC with LLM-MATLOGIC and design a closed-loop diagnostic suite that (i) validates the necessity of scoped enforcement and (ii) attributes failures to the specifier versus the executor, yielding actionable insights beyond end-to-end accuracy.

## 2. Executable Exchange Contract

### 2.1. Motivation: Negation Is Fragile

Negation is fragile over KGs because it depends on *intermediate evidence*: a system must decide **where** an exclusion attaches (which step/branch) and **when** it is enforced (before or after evidence is collapsed). Leaving either implicit admits incompatible executions.

**Timing: witness-level exclusions cannot be recovered post hoc.** Consider "exclude cameo appearances," where an answer entity $a$ is supported by a witness attribute $r$ (e.g., role type). Let Proj drop witness attributes and keep only the answer entities. Two intermediate states may project to the same answer set, yet differ under the intended exclusion:

$$\mathsf{Proj}\big(\{(a, \mathsf{cameo})\}\big) = \mathsf{Proj}\big(\{(a, \mathsf{lead})\}\big) = \{a\},$$

$$\mathsf{Proj}\big(\mathsf{Mask}_\phi(\{(a, \mathsf{cameo})\})\big) = \emptyset \neq \{a\}.$$

Once projection discards witness for $\phi$, no answer-only post-filter can faithfully recover witness-level semantics. Such negation must be enforced *before* projection.

**Scope: exclusions may drift across branches.** For a disjunctive query such as "Movie-A **OR** Movie-B, excluding cameo *witnesses in Movie-A*," the exclusion is intended to constrain only the Movie-A branch. Applying it after branches are merged can over-prune candidates from other branches (scope drift), contradicting the intended reading.

In summary, evidence-dependent negation should be treated as an evidence-level constraint that is both *scope-bound* and enforced *pre-projection*. App. A provides minimal unit tests isolating these two failure modes.

### 2.2. Problem Formulation

We define a minimal EEC between a front-end specifier and a KG executor. The specifier sends a small typed request, and the executor returns answers with a verifiable trace.

Crucially, exclusions are expressed as *scoped predicates* evaluated on *intermediate evidence* (Sec. 2.1).

**Definition 2.1** (Contract-visible Intermediate State)**.** Executing an operator skeleton $\mathcal{T}$ on a KG induces intermediate states. EEC requires each state to expose a minimal evidence interface as a 5-tuple

$$S_t = \langle \texttt{scope\_id}, \mathcal{V}_{\text{out}}^{(t)}, \mathcal{V}_{\text{ev}}^{(t)}, \text{Type}_t, \Pi_t \rangle,$$

where $\texttt{scope\_id}$ identifies the state, $\mathcal{V}_{\text{out}}^{(t)}$ are *output slots*, $\mathcal{V}_{\text{ev}}^{(t)}$ are *evidence slots*, $\text{Type}_t$ maps each slot to a KG type, and $\Pi_t$ is the *candidate evidence space*, a set of typed assignments over $\mathcal{V}_{\text{out}}^{(t)} \dot\cup \mathcal{V}_{\text{ev}}^{(t)}$ (disjoint union).

*Remark.* $\Pi_t$ can be viewed as a typed evidence table: each $z \in \Pi_t$ is a candidate assignment. We use set semantics in the main text.

**Definition 2.2** (Executable Exchange Contract (EEC))**.** An EEC is a request–response interface

$$\text{EEC}_{\text{req}} = \langle \mathcal{T}, E_q, \mathcal{N} \rangle, \qquad \text{EEC}_{\text{resp}} = \langle A_{\text{ans}}, \tau \rangle.$$

Here $\mathcal{T}$ is a normalized operator skeleton that induces a graph of states $\{S_t\}$ when executed on a KG $G$ (App. B), $E_q$ are anchor entities (linked KG IDs), and $\mathcal{N} = \langle \texttt{neg\_flag}, \mathcal{D} \rangle$ specifies exclusions with $\texttt{neg\_flag} = 0 \Rightarrow \mathcal{D} = \emptyset$. Each exclusion is a scoped constraint

$$d = \langle \texttt{scope\_id}, \phi_d \rangle \in \mathcal{D},$$

where $\texttt{scope\_id}$ selects a targeted state $S_t$ and $\phi_d$ is an executable predicate evaluated on candidates in $\Pi_t$ (App. B).

**Request.** The specifier sends $\langle \mathcal{T}, E_q, \mathcal{N} \rangle$: $\mathcal{T}$ defines the intermediate-state topology (including branching/merging), $E_q$ anchors execution to KG entity IDs, and $\mathcal{N}$ encodes scoped exclusions. Enforcing $d = \langle \texttt{scope\_id}, \phi_d \rangle$ removes the candidates in the target $\Pi_t$ that violate $\phi_d$.

**Response.** The executor returns $\langle A_{\text{ans}}, \tau \rangle$, where $A_{\text{ans}}$ assigns the output slots of the final state and $\tau$ supports verification (App. B). Formal guarantees are given in Sec. 2.3.

## 2.3. Contract Guarantees

EEC is implementation-agnostic but enforces three guarantees: *Executable* (exchanged objects are directly runnable), *Scoped* (exclusions bind to the intended intermediate state and are enforced before evidence is discarded), and *Verifiable* (outputs are checkable via minimal traces). Let $S_t = \langle \texttt{scope\_id}, \mathcal{V}_{\text{out}}^{(t)}, \mathcal{V}_{\text{ev}}^{(t)}, \text{Type}_t, \Pi_t \rangle$ be a contract-visible state (Def. 2.1) and $\mathcal{V}_t := \mathcal{V}_{\text{out}}^{(t)} \dot\cup \mathcal{V}_{\text{ev}}^{(t)}$. We use two basic operators on evidence spaces: slot projection $\pi_U(\Pi)$ keeps only slots in $U$ (dropping the rest), and masking $\text{Mask}_\psi(\Pi)$ removes candidates that violate the predicate $\psi$. Full notation is in App. C.

**Requirement 2.3** (C1, Executable)**.** Each state is directly runnable: candidates are well-typed KG objects and all evaluated predicates bind to slots exposed by the state. Formally, for every state $S_t$,

$$\forall z \in \Pi_t, \ \forall v \in \mathcal{V}_t : \ z(v) \in \text{Dom}\big(\text{Type}_t(v)\big),$$
$$\forall \phi \in \Phi_t : \ \text{FV}(\phi) \subseteq \mathcal{V}_t.$$

where $\Phi_t$ includes all predicates evaluated at $S_t$ (in particular, all exclusions targeting $S_t$).

**Requirement 2.4** (C2, Scoped)**.** Each exclusion is a scoped constraint $d = \langle \texttt{scope\_id}, \phi_d \rangle \in \mathcal{D}$ targeting a unique state $S_t$. EEC enforces: (i) $\texttt{scope\_id}$ selects exactly one state $S_t$, and (ii) the predicate fits the state, $\text{FV}(\phi_d) \subseteq \mathcal{V}_t$. If $\phi_d$ depends on evidence slots, i.e., $\text{FV}(\phi_d) \cap \mathcal{V}_{\text{ev}}^{(t)} \neq \emptyset$, then the exclusion must be enforced *before* answer projection:

$$\Pi_t \leftarrow \text{Mask}_{\phi_d}(\Pi_t), \qquad A_t^{(d)} = \pi_{\mathcal{V}_{\text{out}}^{(t)}}(\Pi_t).$$

Multiple exclusions at the same $\texttt{scope\_id}$ are applied by conjunction (equivalently, sequential masking).

**Requirement 2.5** (C3, Verifiable)**.** Executing $\mathcal{T}$ yields a final state with evidence space $\Pi_{\text{final}}$ and designated output slots $\mathcal{V}_{\text{out}}$. Answers are the output-slot projection:

$$A_{\text{ans}} = \pi_{\mathcal{V}_{\text{out}}}(\Pi_{\text{final}}).$$

The trace $\tau$ must make outputs checkable: for each $a \in A_{\text{ans}}$, $\tau$ contains (explicitly or implicitly) at least one witness $z \in \Pi_{\text{final}}$ such that $z|_{\mathcal{V}_{\text{out}}} = a$ and $z$ satisfies all enforced exclusions.

**Summary.** C1 makes contract objects executable, C2 removes ambiguity in negation by binding exclusions to scoped intermediate evidence, and C3 enables auditing by requiring a witness-backed trace for reported answers.

## 2.4. A Canonical Enforcement Boundary for Negation

C2 requires each exclusion to be (i) *scope-bound* to the correct intermediate state ($\texttt{scope\_id}$) and (ii) enforced on intermediate evidence *before* projection discards evidence slots. We show these are not merely design choices: in general, moving an exclusion across projection or across a disjunctive merge changes semantics.

**Two non-commutativities.** Let $\mathcal{X} := \mathcal{V}_{\text{out}}^{(t)}$ be the output slots at state $S_t$, with evidence space $\Pi_t$. Write $N_\phi(\Pi) := \text{Mask}_\phi(\Pi)$ and $P_{\mathcal{X}}(\Pi) := \pi_{\mathcal{X}}(\Pi)$ (Sec. 2.3). **(Projection)** If $\phi_d$ is not output-determined, then masking cannot be simulated by any answer-only post-filter:

$$P_{\mathcal{X}}\big(N_{\phi_d}(\Pi_t)\big) \ \neq \ F\big(P_{\mathcal{X}}(\Pi_t)\big) \text{ for any fixed } F \text{ (in general).}$$

(Propo. D.1 in App. D.) **(Disjunction)** For a disjunctive merge $\Pi = \Pi_1 \cup \Pi_2$, if $d$ is intended to constrain only

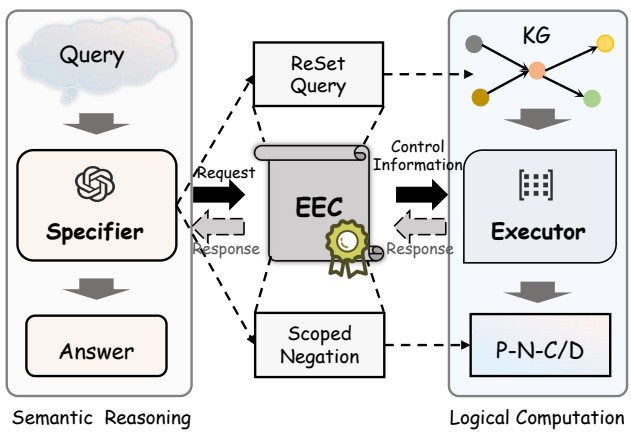

Figure 1. LLM-MATLOGIC framework: semantic specification with KG-based logical execution.

the alternative containing the targeted state (say, $\Pi_1$), then global masking after merging can cause scope drift:

$$N_{\phi_d}(\Pi_1 \cup \Pi_2) \ \neq \ N_{\phi_d}(\Pi_1) \cup \Pi_2 \qquad \text{(in general)}.$$

(Propo. D.2 in App. D.)

**Implication.** Propos. D.1 and D.2 jointly pin down the canonical boundary of a scoped exclusion $d = \langle \text{scope\_id}, \phi_d \rangle$: the mask must be applied (i) *within* the alternative/state selected by scope\_id (preventing scope drift), and (ii) *before* projection discards evidence needed to evaluate $\phi_d$. Consequently, any EEC execution admits an equivalent normalized form in which candidates are first produced (**P**), scoped masks compiled from $\mathcal{D}$ are applied next (**N**), and only then are intermediate results composed via conjunction/disjunction (**C/D**) before the final projection (Cor. D.3 in App. D and Prop. E.1 in App. E). We realize this as **P→N→C/D**, with **N** compiled from $\mathcal{D}$.

## 3. Methodology

We implement EEC as the LLM-MATLOGIC framework (Figure 1), where the LLM outputs a contract request with scoped-negation metadata, and MATLOGIC executes it over the KG using the canonical $P \rightarrow N \rightarrow C/D$ schedule. The full pipeline is shown in App G (Figure 5).

### 3.1. LLM-MATLOGIC: EEC Instantiation

We instantiate the EEC with a front-end *specifier* (LLM) and a matrix *executor* (MATLOGIC). Given a natural-language query $q$ and a KG $G$, the LLM compiles $q$ into a request $R$, and MATLOGIC executes $R$ on $G$ under the canonical **P→N→C/D** schedule to return $\langle A_{\text{ans}}, \tau \rangle$.

**Request.** The LLM produces

$$R = \langle E_q, \ \mathcal{P}, \ \text{SCOPEOF}, \ \mathcal{B} \rangle,$$

---

**Algorithm 1** MATLOGIC executor

**Require:** KG adjacency $W$; hop limit $K$; anchor entity set $E_q$; grouping $\mathcal{P} = \{P_k\}_{k=1}^M$ over candidate paths/atoms $\{p_i\}_{i=1}^N$; a scope map $\text{SCOPEOF} : [N] \rightarrow \mathcal{S}$; scoped exclusions $\mathcal{B} = \{(s, B_s)\}$ with $s \in \mathcal{S}$

**Ensure:** Answer set $A_{\text{ans}}$ and trace $\tau$
1: Construct the $K$-hop closure $W_{\text{Bool}}$ from $W$
2: Construct provenance for witness reconstruction, e.g., $W_{\text{step}}$ and predecessor pointers $W_{\text{pred}}$
3: {N (compile): pre-compile scope-local reachability masks}
4: **for** each $(s, B_s) \in \mathcal{B}$ **do**
5:    $M_s \leftarrow \text{COMPILEMASK}(W_{\text{Bool}}, B_s)$ {pair mask over reachable pairs}
6:    $W_{\neg,s} \leftarrow W_{\text{Bool}} \odot M_s$ {masked closure for scope $s$}
7: **end for**
8: {P → N: produce atomic evidences under the effective (masked) closure of their scope}
9: **for** $k = 1$ to $M$ **do**
10:    **for** each atom/path index $i \in P_k$ **do**
11:       $s \leftarrow \text{SCOPEOF}(i)$
12:       $W_{\text{eff}} \leftarrow W_{\text{Bool}}$
13:       **if** $s \in \text{dom}(W_\neg)$ **then**
14:          $W_{\text{eff}} \leftarrow W_{\neg,s}$
15:       **end if**
16:       $E[i] \leftarrow \text{PRODUCEEVIDENCE}(W_{\text{eff}}, E_q, p_i)$ {Produce uses $W_{\text{eff}}$ (and internal relation filters) to enumerate endpoints}
17:    **end for**
18: **end for**
19: {D: disjunction within each group}
20: **for** $k = 1$ to $M$ **do**
21:    $E[P_k] \leftarrow \emptyset$
22:    **for** each $i \in P_k$ **do**
23:       $E[P_k] \leftarrow \text{UNION}(E[P_k], E[i])$
24:    **end for**
25: **end for**
26: {C: conjunction across groups}
27: $E_{\text{final}} \leftarrow E[P_1]$
28: **for** $k = 2$ to $M$ **do**
29:    $E_{\text{final}} \leftarrow \text{INTERSECT}(E_{\text{final}}, E[P_k])$
30: **end for**
31: $A_{\text{ans}} \leftarrow \text{PROJECT}(E_{\text{final}})$
32: $\tau \leftarrow \text{BUILDTRACE}(A_{\text{ans}}, E, \mathcal{P}, E_q, \text{SCOPEOF}, \mathcal{B}, W_{\text{pred}})$
33: **return** $(A_{\text{ans}}, \tau)$

---

where $E_q$ are anchor entities (linked KG IDs). Conceptually, $\langle \mathcal{P}, \text{SCOPEOF} \rangle$ is a normalized serialization of the operator skeleton $\mathcal{T}$ (App. B), and $\mathcal{B}$ is a grounded instantiation of the exclusion set $\mathcal{D}$ (Sec. 2.2). Specifically: (i) the LLM links entities and outputs $E_q$; (ii) it decomposes $q$ into $N$ atomic evidence-producing items $\{a_i\}_{i=1}^N$ and a grouped composition schema $\mathcal{P} = \{P_k\}_{k=1}^M$, interpreted as $\bigwedge_{k=1}^M \left( \bigvee_{i \in P_k} a_i \right)$ (disjunction within a group, conjunction across groups); (iii) for negation, it outputs a scope map $\text{SCOPEOF} : [N] \rightarrow \mathcal{S}$ and scope-indexed banned-entity sets $\mathcal{B} = \{(s, B_s)\}$, where $B_s \subseteq \{\text{ent\_id}\}$ lists KG entity IDs to exclude within scope $s$. Intuitively, the LLM binds each excluded entity (after linking) to the scope(s) corresponding to the negated clause, yielding $\mathcal{B}$.

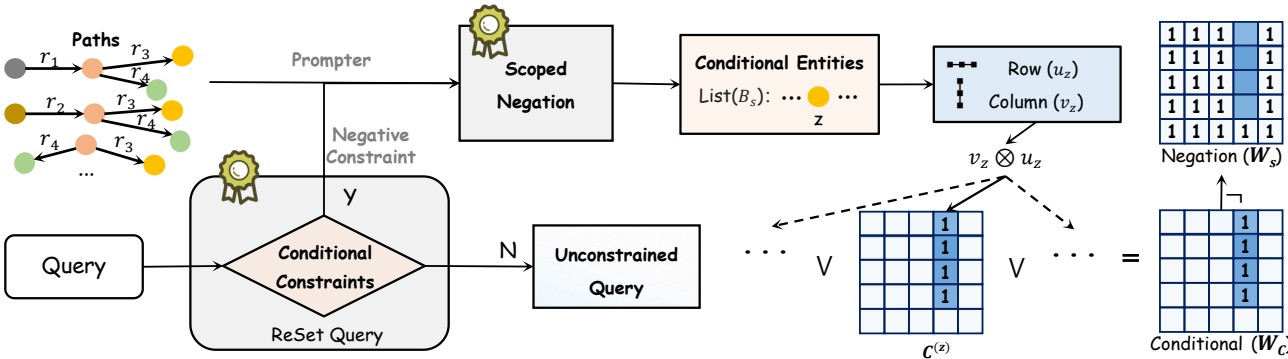

*Figure 2.* **Scoped negation mask construction.** The specifier identifies scoped exclusions as a banned set $B_s$; MatLogic constructs $C^{(z)} = v_z \otimes u_z$ for each $z \in B_s$ and aggregates them to form a scope-local mask used for constrained execution.

**Execution.** Given $R$, MATLOGIC deterministically executes: **P** (*Produce*) compute witness-carrying intermediate evidence for all atomic items from anchors $E_q$; **N** (*Negate*) compile each $B_s$ into a scope-local structural mask and apply it to the corresponding intermediate evidence identified by SCOPEOF; **C/D** (*Compose*) combine intermediate results by disjunction within each group $P_k$ and conjunction across groups, then project the final constrained evidence to obtain $A_{\text{ans}}$. In addition to the answer set, MATLOGIC returns a trace $\tau$ backed by its provenance/step encoding, enabling auditing against the scope-bound exclusions in $\mathcal{B}$.

**Why this order?** By the canonical enforcement boundary (Sec. 2.4) and the normalization result (App. E), evidence-dependent and scope-bound exclusions must be enforced on intermediate evidence within the targeted scope, prior to any merge/projection, without changing semantics. We therefore adopt **P→N→C/D** as the canonical schedule.

### 3.2. MATLOGIC with P–N–C/D Schedule

This section describes how MATLOGIC executes the compiled request $R = \langle E_q, \mathcal{P}, \text{SCOPEOF}, \mathcal{B} \rangle$ to return $\langle A_{\text{ans}}, \tau \rangle$. We evaluate $R$ in a normalized grouped form that exposes exactly the control information needed for execution: (i) the merge structure (disjunction within groups vs. conjunction across groups) and (ii) the scope boundaries where exclusions must be enforced.

**Inputs.** Compilation yields anchors $E_q \subseteq \mathcal{E}$, atomic items $\{a_i\}_{i=1}^{N}$, a grouping $\mathcal{P} = \{P_k\}_{k=1}^{M}$, a scope map $\text{SCOPEOF} : [N] \to \mathcal{S}$, and exclusions $\mathcal{B} = \{(s, B_s)\}$. We interpret $\mathcal{P}$ as the normalized composition $\bigwedge_{k=1}^{M} \left( \bigvee_{i \in P_k} a_i \right)$, i.e., disjunction within a group and conjunction across groups. Each atom $a_i$ is assigned a stable scope $\text{SCOPEOF}(i)$, and an exclusion $(s, B_s)$ is enforced only on the intermediate evidence produced by atoms with $\text{SCOPEOF}(i) = s$ (by normalization, it suffices to index scope at the level of atomic producers). Algorithm 1

summarizes the end-to-end evaluation.

**P: Produce (PRODUCEEVIDENCE).** MATLOGIC converts the KG into a sparse adjacency matrix $W$ and precomputes reachability information up to hop limit $K$. The Boolean $K$-hop closure

$$W_{\text{Bool}} = I \vee W \vee W^2 \vee \cdots \vee W^K$$

supports fast coverage checks. For auditability, we additionally store provenance sufficient to reconstruct a concrete witness. We record the first-reach step matrix

$$(W_{\text{step}})_{ij} = \min\{ k \in \{0, \ldots, K\} : (W^k)_{ij} = 1 \},$$

with $(W_{\text{step}})_{ij} = \infty$ if unreachable, and a corresponding predecessor pointer structure $W_{\text{pred}}$ used by BUILDTRACE, where for any reachable pair $(i, j)$ with $k = (W_{\text{step}})_{ij} < \infty$, $W_{\text{pred}}[i, j]$ stores one predecessor $p$ on a shortest witness such that $(W_{\text{step}})_{ip} = k - 1$ and $W_{pj} = 1$.

Given anchors $E_q$ (KG entity IDs) and an atomic item $a_i$ (a candidate path specification), PRODUCEEVIDENCE returns an intermediate evidence vector $E_i$ (a sparse indicator over KG entities) via $K$-hop propagation:

$$x^{(0)} = \mathbf{1}_{E_q},$$

$$x^{(k)} = x^{(k-1)}W \ (k = 1, \ldots, K), \qquad E_i := \bigvee_{k=0}^{K} x^{(k)}.$$

(When an atomic item restricts relations/directions, PRODUCEEVIDENCE applies relation-filtered transitions internally; we write $W$ for simplicity.) Intuitively, $E_i$ represents the set of candidate endpoints for item $a_i$, while witness paths are recoverable from provenance (e.g., $W_{\text{pred}}$ together with the chosen supporting items).

**N: Negate (COMPILEMASK, APPLYMASK).** Exclusions are given as scope-indexed banned-entity sets $\mathcal{B} = \{(s, B_s)\}$, where $B_s$ lists KG entity IDs to exclude *within scope $s$*. MATLOGIC compiles a scope-local reachability

mask once per scope and applies it to all atomic items with $\textsc{ScopeOf}(i) = s$, as illustrated in Figure 2. For a banned entity $z$, $\textsc{CompileMask}$ blocks all pairs $(u, v)$ for which a $K$-hop witness can traverse $z$, i.e., $u \rightsquigarrow z$ and $z \rightsquigarrow v$ within $K$ hops. Let $\mathrm{pre}(z) = W_{\mathrm{Bool}}[:, z]$ and $\mathrm{suf}(z) = W_{\mathrm{Bool}}[z, :]$ denote $K$-hop reachability *to* and *from* $z$. The blocked region induced by $z$ is $C^{(z)} = \mathrm{pre}(z) \otimes \mathrm{suf}(z)$, and aggregating over a banned set yields scope-local mask $M_s = \neg \bigvee_{z \in B_s} C^{(z)}$. Rather than post-hoc filtering endpoints, $\textsc{ApplyMask}$ enforces the exclusion by restricting propagation/closure inside the scope:

$$W_{\neg, s} = W_{\mathrm{Bool}} \odot M_s,$$

and computing (or updating) the intermediate evidence for atoms in scope $s$ using $W_{\neg, s}$ in place of $W_{\mathrm{Bool}}$ (equivalently, disallowing any witness that traverses a banned entity). This enforcement is *scope-local* and occurs *before* any merge or projection, matching the EEC boundary.

**C/D: Compose ($\textsc{Union}$, $\textsc{Intersect}$, $\textsc{Project}$, $\textsc{Build-Trace}$).** After P and N, we compose intermediate evidences using the grouped schema $\mathcal{P} = \{P_k\}_{k=1}^M$. Disjunction *within* each group is $E_{P_k} = \bigvee_{i \in P_k} E_i$ ($\textsc{Union}$), and conjunction *across* groups yields the final constrained evidence $E_\star = \bigwedge_{k=1}^M E_{P_k}$ ($\textsc{Intersect}$). $\textsc{Project}$ returns answers as entity IDs:

$$A_{\mathrm{ans}} = \{ e \in \mathcal{E} : (E_\star)_e = 1 \}.$$

Finally, $\textsc{BuildTrace}$ constructs the contract trace $\tau$ so that each reported answer admits a concrete witness. For each $e \in A_{\mathrm{ans}}$, $\tau$ records (i) one supporting atomic item index $i_k \in P_k$ for every group $k$ such that $(E_{i_k})_e = 1$, and (ii) the predecessor pointers needed to reconstruct a $K$-hop witness for each chosen atom. Full trace detail in App. B. This enables recovering, on demand, a concrete witness for each returned answer while auditing scope-bound exclusions.

### 3.3. Closing the Loop with the EEC Contract

We expose only contract-relevant control metadata (merge structure and scopes) and execute under $\mathbf{P} \rightarrow \mathbf{N} \rightarrow \mathbf{C/D}$ boundary. By the enforcement boundary and normalization, exclusions must be enforced on intermediate evidence within the targeted scope, prior to any merge or projection.

**EEC compliance (C1–C3).** *Executable (C1).* All exchanged objects are KG-grounded (entity IDs and KG-indexed sparse structures), and the executor uses only directly executable matrix/element-wise operations. *Scoped (C2).* Each exclusion $(s, B_s) \in \mathcal{B}$ is enforced only on intermediate evidence produced in scope $s$ (via $\textsc{ScopeOf}$), and is applied before any cross-scope composition or projection. *Verifiable (C3).* $\textsc{MatLogic}$ returns a trace $\tau$ backed by its provenance encoding (e.g., predecessor pointers together

with supporting atomic items), enabling recovery of a concrete witness for each reported answer and auditing against the enforced exclusions.

## 4. Experiment

### 4.1. Experimental Setup

We evaluate EEC on two complementary benchmarks: structured CQA (to test executor-side scoped enforcement under unambiguous query semantics) and end-to-end NL-KGQA (to test specifier-side compilation of scope-bound exclusions from underspecified text).

**Datasets.** For CQA, we use FB15k-237 (Toutanova et al., 2015) and NELL995 (Carlson et al., 2010), covering 14 query patterns (9 standard CQA and 5 negation-aware patterns). Natural-language questions are generated from the structured templates following LARK-style question construction (Choudhary & Reddy, 2023). For NL-KGQA, we use CWQ (Talmor & Berant, 2018) and WebQSP (Yih et al., 2016) over Freebase. For each query/question, we retrieve an entity-centric $K$-hop subgraph around the query anchors with a fixed budget (details in App. H.4); all methods operate on the same retrieved subgraphs for fairness. For the main experiments in Section 4.2, we filter out simple subgraphs and randomly sample the same number of queries for each query pattern. For diagnostics in Section 4.3, we use a smaller subset with 200 queries per pattern (2,800 total). NL-KGQA evaluation includes 1,000 questions.

**Baselines.** On CQA, we compare against representative embedding-based CQA methods (e.g., GQE, Query2Box, BetaE, CQD-Hybrid) and prompting-based baselines (LARK, MFC). On NL-KGQA, we compare against ToG and MFC. Full details are in App. H.3.

**Metrics and implementation.** For CQA, we report MRR using a fixed structure-driven ranking to map MatLogic's reachable candidates to a ranked list (details in App. H.5). For NL-KGQA, we report Hits@1 and the average number of LLM calls per question. We use GPT-3.5 as the specifier (temperature=0, single-pass decoding; prompt in App. H.4).

### 4.2. Main Results

We report results on two complementary CQA suites and on end-to-end NL-KGQA. For CQA, Table 1 summarizes MRR on nine standard query patterns without explicit negation (1p/2p/3p/2i/3i/ip/pi/2u/up), while Table 2 reports MRR on five negation-aware patterns (2in/3in/inp/pin/pni), which stress-test exclusion handling in composed queries. [1] For NL-KGQA, Table 3 reports Hits@1 together with the average number of LLM calls, reflecting the accuracy–cost

---

[1] Notation: p=path, i=intersection, u=union, n=negation. Full definitions are in the appendix.

*Table 1.* MRR Comparison of LLM-MATLOGIC and Two Interaction Methods in Nine Query Patterns: Embedding and Prompting

| Interaction Methods | Models | 1p | 2p | 3p | 2i | 3i | ip | pi | 2u | up |
|---|---|---|---|---|---|---|---|---|---|---|
| **Dataset: FB15K-237** | | | | | | | | | | |
| Euclidean Geometry | GQE | 34.6 | 8.3 | 4.7 | 22.9 | 33.1 | 16.8 | 11.2 | 8.1 | 6.5 |
| | Query2Box | 41.2 | 10.1 | 5.8 | 28.6 | 42.1 | 22.0 | 13.6 | 11.6 | 8.3 |
| | BetaE | 38.7 | 12.5 | 10.3 | 27.5 | 42.7 | 23.1 | 14.1 | 12.9 | 10.6 |
| Non-Euclidean Geometry | HQE | 38.1 | 20.8 | 16.4 | 25.8 | 34.9 | 17.7 | 9.5 | 15.0 | 17.3 |
| | HypE | 47.5 | 35.2 | 23.2 | 33.2 | 43.1 | 18.2 | 30.9 | 42.3 | **26.4** |
| Query Decomposition | CQD | 45.3 | 11.6 | 8.4 | 31.8 | 42.5 | 26.9 | 16.4 | 13.7 | 5.2 |
| | CQD-Hybrid | 50.7 | 18.9 | 13.5 | 34.4 | **45.6** | 30.5 | 21.3 | 15.8 | 7.8 |
| LLM+Prompting | LARK | 65.6 | 34.7 | 21.8 | 39.7 | 39.3 | 21.4 | **42.8** | 47.4 | 23.5 |
| | MFC | 67.9 | 35.2 | 30.9 | 42.4 | 43.6 | 27.6 | 37.7 | **48.2** | 20.1 |
| EEC | LLM-MATLOGIC | **69.2** | **58.2** | **42.5** | **57.1** | 43.2 | **31.8** | 24.2 | 39.6 | 26.3 |
| **Dataset: NELL995** | | | | | | | | | | |
| Euclidean Geometry | GQE | 33.3 | 12.5 | 9.4 | 28.3 | 34.1 | 18.9 | 14.2 | 9.2 | 8.4 |
| | Query2Box | 43.5 | 14.2 | 10.6 | 34.8 | 44.0 | 22.1 | 16.2 | 12.3 | 10.8 |
| | BetaE | 54.1 | 13.8 | 10.9 | 38.5 | 46.7 | 25.2 | 14.4 | 13.1 | 8.9 |
| Non-Euclidean Geometry | HQE | 36.4 | 21.5 | 18.8 | 24.1 | 35.9 | 10.6 | 13.9 | 21.0 | 16.2 |
| | HypE | 47.2 | 32.9 | 27.5 | 34.3 | 47.6 | 32.5 | 13.1 | 21.6 | 26.7 |
| Query Decomposition | CQD | 51.7 | 19.6 | 13.2 | 39.7 | 48.5 | 29.8 | 21.5 | 16.8 | 9.4 |
| | CQD-Hybrid | 58.9 | 27.7 | 19.1 | 45.4 | **56.3** | 31.3 | 24.7 | 16.2 | 10.5 |
| LLM+Prompting | LARK | 73.2 | 38.3 | 25.5 | 46.1 | 42.2 | 23.5 | 26.8 | 26.9 | **36.7** |
| | MFC | 75.4 | 39.6 | 32.9 | 49.5 | 46.8 | 28.6 | **28.3** | **33.1** | 31.6 |
| EEC | LLM-MATLOGIC | **79.4** | **64.5** | **44.2** | **59.7** | 44.5 | **31.9** | 22.7 | 24.4 | 35.2 |

*Table 2.* MRR comparison on five negation-aware patterns.

| Model | 2in | 3in | inp | pin | pni |
|---|---|---|---|---|---|
| **Dataset: FB15k-237** | | | | | |
| BetaE | 5.9 | 8.2 | 7.9 | 4.1 | 2.8 |
| LARK | 7.5 | 4.8 | **23.3** | **18.7** | 3.6 |
| LLM-MATLOGIC | **11.4** | **12.7** | 15.1 | 7.5 | **10.9** |
| **Dataset: NELL995** | | | | | |
| BetaE | 5.7 | 7.5 | 9.6 | 3.3 | 3.1 |
| LARK | 10.6 | 6.9 | **24.4** | **14.5** | 7.8 |
| LLM-MATLOGIC | **13.8** | **15.2** | 17.9 | 6.7 | **13.3** |

*Table 3.* KGQA performance (Hits@1) and average LLM calls.

| Model | CWQ | | WebQSP | |
|---|---|---|---|---|
| | Hits@1 | Calls | Hits@1 | Calls |
| **Fine-tuned** | | | | |
| DiFaR[2] | – | 0 | 65.3 | 0 |
| DecAF | 70.4 | 0 | 82.1 | 0 |
| **Prompting** | | | | |
| IO-prompting | 45.2 | 1 | 67.2 | 1 |
| CoT-prompting | 47.0 | 1 | 67.7 | 1 |
| **LLM+KG** | | | | |
| ToG | 51.8 | 14.3 | 72.2 | 10.7 |
| MFC | 62.8 | 9.7 | **78.9** | 7.5 |
| LLM-MATLOGIC | **63.5** | **3.0** | 76.7 | **3.0** |

tradeoff of the specifier–executor protocol.

**CQA: strong compositional reasoning.** In Table 1, LLM-MATLOGIC achieves the best average performance across standard CQA patterns on both FB15k-237 and NELL995. The advantage is most visible on composition-heavy queries (e.g., 2p/2i): on FB15k-237 it reaches 58.2/57.1 versus the best prompting baseline at 35.2/42.4, and on NELL995 it reaches 64.5/59.7 versus 39.6/49.5.

**CQA: negation-aware patterns.** Table 2 shows larger gains on negation-aware patterns, where LLM-MATLOGIC outperforms strong prompting and embedding baselines. LLM-MATLOGIC is strongest on 2in/3in and pni (FB15k-237: 11.4/12.7/10.9 vs. LARK 7.5/4.8/3.6; NELL995:

13.8/15.2/13.3 vs. 10.6/6.9/7.8). On inp/pin, LARK remains higher, indicating that some mixed compositions remain challenging; we analyze this gap with contract-aligned diagnostics in Sec. 4.3.

**NL-KGQA: accuracy-cost tradeoff.** In Table 3, LLM-MATLOGIC attains the best Hits@1 on CWQ with fewer LLM calls than multi-round LLM+KG baselines, outperforming ToG (51.8 with 14.3 calls) and matching MFC's accuracy (62.8) at substantially lower cost (9.7 calls). On WebQSP, it remains competitive (76.7) with 3.0 calls, com-

Table 4. Performance with different LLM specifiers across 14 query patterns.

| Method | 1p | 2p | 3p | 2i | 3i | ip | pi | 2u | up | 2in | 3in | inp | pin | pni |
|---|---|---|---|---|---|---|---|---|---|---|---|---|---|---|
| DeepSeek-R1-1.5B | 62.8 | 53.1 | 35.3 | 43.4 | 34.3 | 20.8 | 17.5 | 22.8 | 18.2 | 2.4 | 1.8 | 3.4 | 2.6 | 2.1 |
| Llama-3-8B | 67.5 | 55.5 | 38.1 | 50.8 | 35.6 | 30.4 | 23.7 | 34.8 | 23.1 | 5.5 | 4.9 | 8.7 | 7.1 | 6.5 |
| GPT-3.5-turbo | **69.2** | **58.2** | **42.5** | **57.1** | **43.2** | **31.8** | **24.2** | **39.6** | **26.3** | **11.4** | **12.7** | **15.1** | **7.5** | **10.9** |

Table 5. Results under unknown/known constraints.

| Type | Unknown-constraints | | Known-constraints | |
|---|---|---|---|---|
| | MRR | Hits@1 | MRR | Hits@1 |
| 2in | 0.162 | 0.216 | **0.479** | **0.392** |
| 3in | 0.187 | 0.136 | **0.278** | **0.186** |
| inp | 0.238 | 0.204 | **0.385** | **0.341** |
| pin | 0.213 | 0.195 | **0.297** | **0.251** |
| pni | 0.197 | 0.155 | **0.407** | **0.328** |

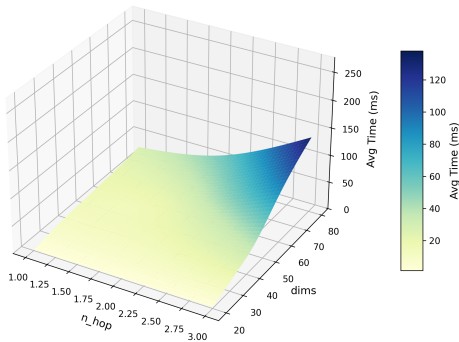

Figure 3. The impact of structure on MatLogic computation time.

pared to MFC (78.9 with 7.5 calls) and ToG (72.2 with 10.7 calls), highlighting the benefit of a single structured EEC request over multi-round prompting.

### 4.3. Contract-aligned Diagnostics

Main results show that LLM-MATLOGIC performs strongly on both CQA and NL-KGQA. We next ask three diagnostic questions to close the loop with our contract view.

**Q1: If the exclusion constraints are correct, can the executor enforce them reliably?** Table 5 compares performance under *unknown* constraints (generated by the specifier) versus *known* constraints (oracle negation constraints) while keeping the executor fixed. Across five negation-aware patterns, known constraints yield a large improvement (avg. MRR: 0.199→0.369; avg. Hits@1: 0.181→0.300), with the largest gains on evidence-sensitive patterns such as 2in (MRR: 0.162→0.479) and pni (MRR: 0.197→0.407). This indicates that, when binding decisions are correct, the executor can enforce scoped exclusions effectively; a substantial portion of the remaining error stems from contract compilation rather than enforcement.

**Q2: Is performance primarily bottlenecked by specifier?** To isolate specifier effects, Table 4 varies only the LLM specifier while keeping the same EEC protocol and the same MATLOGIC executor. Performance improves consistently with stronger specifiers, and the sensitivity is especially pronounced on negation-aware patterns (e.g., 3in: 1.8→4.9→12.7; 2in: 2.4→5.5→11.4). This supports the separation-of-concerns promised by EEC: exclusion binding/constraint generation is a plug-in compilation component, and stronger specifiers translate directly into better exclusion handling under a fixed executor.

**Q3: Is witness-level scoped enforcement practical?** Figure 3 reports MATLOGIC computation time as a function

of subgraph structure (number of hops and representation dimension). Compute time increases with both hop count and dimensionality, remaining in the sub-second (typically millisecond-to-hundreds-of-milliseconds) regime within our tested range. This suggests that enforcing scoped exclusions on intermediate witnesses is not only semantically motivated but also practical under realistic subgraph budgets.

**Insight.** Overall, the diagnostics indicate that exclusion-rich negation in LLM-to-KG reasoning is primarily a *binding/compilation* bottleneck: once scope-bound constraints are made explicit, enforcement becomes a stable and efficient executor responsibility.

## 5. Related Work

**Formal complex query answering over KGs.** CQA typically assumes an *explicit* logical query (query graph/FOL expression) as input, so the scope of conjunction/disjunction/negation is fixed by the query structure. One family executes logical operators via continuous *query embeddings*, using geometric operations to approximate set intersection/union/complement (Hamilton et al., 2018; Ren et al., 2020; Ren & Leskovec, 2020). Other lines learn reasoning procedures through differentiable rule induction or local subgraph reasoning, sometimes combining symbolic matching with neural scoring to improve coverage on multi-hop and union queries (Yang et al., 2017; Teru et al., 2020; Gregucci et al., 2024). Recent neural-symbolic variants further execute explicit FOL or query-graph structures by combining neural link predictors, one-hop inference, or fuzzy-logic aggregation, rather than relying solely on a single geometric query embedding (Zhu et al., 2022; Zihao

et al., 2023; Yin et al., 2024). Negation is thus well-defined once the query is formalized; our focus instead is on exclusions that originate from natural language, where the *where-to-bind* and *when-to-enforce* are not given upfront.

**LLM-based natural-language KGQA.** Recent systems leverage LLMs either by synthesizing executable programs (e.g., SPARQL) from text (Xie et al., 2022; Li et al., 2024), often with execution feedback, or by iterative retrieval-and-reasoning over KG subgraphs/triples (Sun et al., 2024; Luo et al., 2024; Zhang et al., 2025). While effective, correctly binding exclusions from language remains challenging: it may be unclear which branch an exclusion should constrain and at what stage it should be enforced, especially under branching and composition when negation depends on intermediate evidence. This motivates making scope-bound exclusions explicit *as part of the specifier–executor exchange* and enforcing them at a semantics-preserving boundary.

## 6. Conclusion

Exclusions in real KG queries are often *evidence-dependent*: they constrain how an answer is supported rather than which answer entity appears at the end. Our key takeaway is that handling such negation reliably is less about adding new logical operators and more about making two decisions explicit and enforceable: *where* an exclusion binds in the intermediate computation and *when* it is enforced before witnesses are lost or scopes are entangled. EEC turns these underspecified choices in natural language into interface-level commitments between a specifier and an executor, and MATLOGIC demonstrates that these commitments can be realized by a concrete, efficient execution schedule for scoped negation. More broadly, our diagnostic experiments show that the contract perspective yields actionable insight beyond end-to-end accuracy: it cleanly attributes failures to the *specifier* (scope/set compilation) versus the *executor* (enforcement boundary). Moreover, the same enforcement boundary is consistently necessary across both CQA and end-to-end KGQA, supporting an interface-first place to represent and execute scope-bound exclusions.

## Acknowledgements

This work was supported by National Science and Technology Major Project (2025ZD0123700-5) and the National Natural Science Foundation of China (No.62406144, No. U22B2021 and No. 62272025).

## Impact Statement

This paper studies exclusion-rich negation in LLM-to-KG reasoning and proposes an exchange interface (EEC) with a matrix-based executor (MATLOGIC) to make exclusion binding and enforcement explicit and executable. A potential positive impact is improved reliability of KG-backed query systems that must honor "not" conditions, reducing cases where returned answers appear plausible but violate the intended exclusion. By separating contract compilation (specifier) from contract enforcement (executor), the approach also supports clearer testing and debugging in practical pipelines and can reduce interaction overhead compared to multi-round prompting.

The main limitations and risks follow from specification quality and KG coverage. When an exclusion is linguistically ambiguous, a front-end model may bind it to an unintended scope, which can cause over- or under-exclusion; high-stakes deployments may therefore require additional safeguards beyond the scope of this work. As with other KG-backed systems, results are constrained by incompleteness or bias in the underlying graph, so faithful execution does not guarantee exhaustive or fully representative answers. We view EEC as a step toward more principled interfaces for exclusion handling; future work can integrate stronger compilation strategies and application-level controls.

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

# A. Minimal Unit Tests for Negation

This appendix provides minimal, executable-style unit tests that isolate two independent failure modes of negation over intermediate evidence: **timing** (post-hoc answer filtering is under-specified for witness-level exclusions) and **scope** (global negation can drift across disjunctive branches). The fixture corresponds to the natural-language pattern "Movie-A **OR** Movie-B, excluding cameo **witnesses** in Movie-A". We model intermediate results as witness tuples $(a, r)$, where $a$ is an answer entity (e.g., actor) and $r$ is a witness attribute (e.g., role type). Let $\phi(r) \triangleq [r = \text{cameo}]$ be a *violation predicate*. We use two operators: (i) witness masking $\text{Mask}_\phi$ that removes forbidden witnesses, and (ii) projection $\text{Proj}$ that collapses witness attributes and returns answer entities only.

**Test 1 (Timing): post-hoc filtering cannot recover witness-level semantics.** The key phenomenon is that projection discards the evidence needed to evaluate the exclusion. Algorithm 2 constructs two intermediate states $S^{(1)}$ and $S^{(2)}$ that yield the *same* projected answer set, yet differ under the intended witness-level exclusion. This implies that no answer-only rule can implement the intended semantics:

**Lemma A.1** (No answer-only post-hoc rule). *If* $\text{Proj}(S^{(1)}) = \text{Proj}(S^{(2)})$ *but* $\text{Proj}(\text{Mask}_\phi(S^{(1)})) \neq \text{Proj}(\text{Mask}_\phi(S^{(2)}))$, *then there does not exist a function $g$ such that $g(\text{Proj}(S)) = \text{Proj}(\text{Mask}_\phi(S))$ for all $S$.*

*Proof.* The premise forces $g(\text{Proj}(S^{(1)})) = g(\text{Proj}(S^{(2)}))$, but the right-hand sides differ, a contradiction. $\square$

**Test 2 (Scope): global negation drifts across disjunction.** Negation must also be bound to the correct intermediate branch/state. Algorithm 2 constructs a disjunction with two branches: $B_1$ (Movie-A) and $B_2$ (Movie-B). The intended reading excludes cameo *witnesses in Movie-A* only. Applying $\text{Mask}_\phi$ *globally* to $B_1 \cup B_2$ can therefore over-prune candidates originating from the other branch (scope drift), leading to an answer set that contradicts the intended semantics. By contrast, applying the exclusion *scoped* to the Movie-A branch before merging branches yields the expected result.

These unit tests motivate treating negation as an *evidence-level constraint* that is (i) *scope-bound* to the correct intermediate state/branch and (ii) enforced *pre-projection*. These are precisely the conditions enforced by the EEC contract (Sec. 2) and respected by the canonical execution schedule (Sec. 3).

---

**Algorithm 2** Minimal unit tests for negation (fixture: "Movie-A OR Movie-B, excluding cameo **witnesses** in Movie-A").

---

**Require:** Witness tuples $(a, r)$ where $a$ is an entity (actor) and $r$ is a witness attribute (role); predicate $\phi(r) \triangleq [r = \text{cameo}]$.
**Ensure:** Two unit tests showing why negation must be (i) scope-bound and (ii) enforced pre-projection.
 1: **Operators (defined).**
 2: $\text{Mask}_\phi(S) \triangleq \{(a, r) \in S : \neg\phi(r)\}$ {remove forbidden witnesses}
 3: $\text{Proj}(S) \triangleq \{a : \exists r, (a, r) \in S\}$ {project to answer entities}
 4:
 5: **Test 1 (Timing): post-hoc filtering is insufficient.**
 6: $S^{(1)} \leftarrow \{(\text{Alice}, \text{cameo})\}; \quad S^{(2)} \leftarrow \{(\text{Alice}, \text{lead})\}$
 7: $A_1^{raw} \leftarrow \text{Proj}(S^{(1)}); \quad A_2^{raw} \leftarrow \text{Proj}(S^{(2)})$ {both are $\{\text{Alice}\}$}
 8: $A_1^\star \leftarrow \text{Proj}(\text{Mask}_\phi(S^{(1)})); \quad A_2^\star \leftarrow \text{Proj}(\text{Mask}_\phi(S^{(2)}))$ {$\emptyset$ vs $\{\text{Alice}\}$}
 9: **assert**$(A_1^{raw} = A_2^{raw} \text{ and } A_1^\star \neq A_2^\star)$ {no function of answers alone can match witness-level semantics}
10:
11: **Test 2 (Scope): global negation drifts across branches.**
12: $B_1 \leftarrow \{(\text{Alice}, \text{cameo}), (\text{Alice}, \text{lead})\}$ {Branch 1 (Movie-A)}
13: $B_2 \leftarrow \{(\text{Bob}, \text{cameo})\}$ {Branch 2 (Movie-B)}
14: $A^\star \leftarrow \text{Proj}(\text{Mask}_\phi(B_1) \cup B_2)$ {expected: $\{\text{Alice}, \text{Bob}\}$}
15: $A_2 \leftarrow \text{Proj}(\text{Mask}_\phi(B_1 \cup B_2))$ {global negation (scope drift)}
16: **assert**$(A_2 \neq A^\star)$ {FAIL (scope)}

---

# B. Contract Semantics for EEC

**Skeleton, states, and scopes.** The EEC request contains an operator skeleton $\mathcal{T}$ which, when executed on a KG, induces contract-visible states $S_t = \langle \text{scope\_id}(t), \mathcal{V}_{\text{out}}^{(t)}, \mathcal{V}_{\text{ev}}^{(t)}, \text{Type}_t, \Pi_t \rangle$ (Def. 2.1). An exclusion is a scoped constraint $d = \langle \text{scope\_id}, \phi_d \rangle$ (Def. 2.2), meaning that $\phi_d$ is enforced on candidates in the unique state selected by $\text{scope\_id}$, preventing scope drift across alternatives. In our instantiation, the executor consumes $\mathcal{T}$ through a normalized serialization

$\text{Serialize}(\mathcal{T}) = \langle \mathcal{P}, \text{SCOPEOF} \rangle$, where $\mathcal{P} = \{P_k\}_{k=1}^M$ encodes disjunction-within-group and conjunction-across-groups, and $\text{SCOPEOF} : [N] \to \mathcal{S}$ assigns each atomic evidence producer a stable scope identifier.

**Evidence spaces, masking, and projection.** At state $S_t$, the evidence space $\Pi_t$ is a set (or weighted multiset) of typed candidates $z$ assigning values to slots in $\mathcal{V}_{\text{out}}^{(t)} \dot{\cup} \mathcal{V}_{\text{ev}}^{(t)}$. For a slot subset $U$, let $z|_U$ be restriction and $\pi_U(\Pi_t) = \{z|_U : z \in \Pi_t\}$ be the induced projection. Given an executable predicate $\psi$ with $\text{FV}(\psi) \subseteq \mathcal{V}_{\text{out}}^{(t)} \dot{\cup} \mathcal{V}_{\text{ev}}^{(t)}$, masking removes (or downweights) violating candidates: $\text{Mask}_\psi(\Pi_t) = \{z \in \Pi_t : \psi(z) = 0\}$ under set semantics (weighted variants reweight analogously). Answer projection is $A_t = \pi_{\mathcal{V}_{\text{out}}^{(t)}}(\Pi_t)$. If an exclusion depends on evidence slots (not output-determined), it must be enforced *before* projection; scoped exclusions must be enforced within the targeted alternative/state before disjunctive merging (Sec. 2.4).

**Grounded predicates and the exclusion class used in this paper.** EEC assumes exclusions are grounded (executable without natural-language interpretation). In this paper, exclusions are instantiated as scope-indexed banned-entity sets $\mathcal{B} = \{(s, B_s)\}$: the predicate $\phi_d$ is compiled into a KG-level condition that forbids witnesses traversing any entity in $B_s$ within scope $s$ (Sec. 3.2). Enforcement is scope-local and occurs prior to any merge or projection.

**Trace schema and verifiability (C3).** The response returns $(A_{\text{ans}}, \tau)$, where $\tau$ must certify that each $e \in A_{\text{ans}}$ admits at least one surviving witness. For the grouped normal form $\bigwedge_{k=1}^M (\bigvee_{i \in P_k} a_i)$, we use an implicit trace

$$\tau(e) = \Big\langle \{(k, i_k, s_k)\}_{k=1}^M, \ \{(k, u_k, \ell_k)\}_{k=1}^M \Big\rangle,$$

where $i_k \in P_k$ is one supporting atom with $e$ surviving in its post-mask evidence, $s_k = \text{SCOPEOF}(i_k)$, and $(u_k, \ell_k)$ are witness pointers (e.g., an anchor and hop length). The executor maintains predecessor pointers sufficient to reconstruct one concrete witness per pointer (e.g., a scope-indexed $W_{\text{pred}}^{(s)}$ or an equivalent store), so that reconstructed witnesses respect the enforced exclusions for scope $s_k$.

**Alignment with LLM-MATLOGIC.** The abstract request $\langle \mathcal{T}, E_q, \mathcal{N} \rangle$ is realized as $R = \langle E_q, \mathcal{P}, \text{SCOPEOF}, \mathcal{B} \rangle$, with $\langle \mathcal{P}, \text{SCOPEOF} \rangle = \text{Serialize}(\mathcal{T})$ and $\mathcal{B}$ grounding $\mathcal{D}$. Contract evidence spaces $\Pi_t$ are represented by KG-indexed sparse evidences (e.g., endpoint indicators $E_i$) together with provenance used by $\tau$; contract masking $\text{Mask}_\phi$ is realized by scope-local witness restriction (e.g., via $W_{\neg,s}$-restricted generation) before any composition; final answers are obtained by composing evidences according to $\mathcal{P}$ and projecting to entity IDs to form $A_{\text{ans}}$, with $\tau$ providing auditable witnesses.

## C. Notation for Evidence Spaces and Predicates

We fix notation used by the EEC contract (Sec. 2.3) and the formal results in App. D and E; semantic definitions are given in App. B. Let $S_t = \langle \texttt{scope\_id}, \mathcal{V}_{\text{out}}^{(t)}, \mathcal{V}_{\text{ev}}^{(t)}, \text{Type}_t, \Pi_t \rangle$ be a contract-visible state (Def. 2.1), and let $\mathcal{V}_t := \mathcal{V}_{\text{out}}^{(t)} \dot{\cup} \mathcal{V}_{\text{ev}}^{(t)}$ be its full slot set (disjoint union). A candidate $z \in \Pi_t$ is a typed assignment over $\mathcal{V}_t$. For any $U \subseteq \mathcal{V}_t$, $z|_U$ denotes restriction to slots in $U$, and the slot projection is

$$\pi_U(\Pi_t) := \{ z|_U : z \in \Pi_t \},$$

with answer projection given by $U = \mathcal{V}_{\text{out}}^{(t)}$.

A predicate $\phi$ is an executable boolean test on candidates in $\Pi_t$ and $\text{FV}(\psi) \subseteq \mathcal{V}_t$ denotes the referenced slot set. Masking is written as

$$\text{Mask}_\psi(\Pi) := \{ z \in \Pi : \psi(z) = 0 \}, \qquad \text{and} \quad N_\psi(\Pi) := \text{Mask}_\psi(\Pi).$$

For interface-aligned disjunctive alternatives we write $D(\Pi_1, \Pi_2) := \Pi_1 \cup \Pi_2$. Finally, letting $\mathcal{X} := \mathcal{V}_{\text{out}}^{(t)}$, a predicate $\phi$ is *output-determined* (w.r.t. $\mathcal{X}$) if $\forall z_1, z_2, \ z_1|_{\mathcal{X}} = z_2|_{\mathcal{X}} \Rightarrow \phi(z_1) = \phi(z_2)$ (equivalently, $\phi(z)$ depends only on $z|_{\mathcal{X}}$).

## D. Formal Properties of Exclusion Placement (Sec. 2.4)

We follow Sec. 2.2–2.3 under *set semantics* (masking removes candidates; disjunction merges by union). Fix a contract-visible state $S_t$ with output slots $\mathcal{X} := \mathcal{V}_{\text{out}}^{(t)}$ and evidence slots $\mathcal{Y} := \mathcal{V}_{\text{ev}}^{(t)}$, and let $\mathcal{V}_t = \mathcal{X} \dot{\cup} \mathcal{Y}$. Its evidence space $\Pi$ is a set of candidates (typed assignments) $z$ over $\mathcal{V}_t$. For $U \subseteq \mathcal{V}_t$, write slot projection

$$P_U(\Pi) := \{ z|_U : z \in \Pi \},$$

and for a predicate $\phi$ on candidates, masking

$$N_\phi(\Pi) := \{\, z \in \Pi :\ \phi(z) = 0 \,\}.$$

For interface-aligned disjunctive alternatives, the merge operator is $D(\Pi_1, \Pi_2) := \Pi_1 \cup \Pi_2$. (Weighted variants follow by replacing set membership with support/zero-weight semantics.) A predicate $\phi$ is *output-determined* w.r.t. $\mathcal{X}$ if

$$\forall z_1, z_2 :\ z_1|_{\mathcal{X}} = z_2|_{\mathcal{X}} \implies \phi(z_1) = \phi(z_2). \tag{1}$$

Equivalently, there exists $\tilde{\phi}$ on output assignments such that $\phi(z) = \tilde{\phi}(z|_{\mathcal{X}})$ for all $z$.

**Proposition D.1** (Post-filterability iff output-determined)**.** *Let $\phi$ be a predicate on candidates over $\mathcal{V}_t = \mathcal{X} \dot\cup \mathcal{Y}$. There exists an* answer-only *operator $F$ on projected spaces such that for all evidence spaces $\Pi$,*

$$P_{\mathcal{X}}\big(N_\phi(\Pi)\big) \;=\; F\big(P_{\mathcal{X}}(\Pi)\big) \tag{2}$$

*if and only if $\phi$ is output-determined (Eq. (1)).*

*Proof.* ($\Rightarrow$) Assume (2) holds. Take any two candidates $z_1, z_2$ with the same output assignment $z_1|_{\mathcal{X}} = z_2|_{\mathcal{X}} =: x$. Let $\Pi^{(1)} = \{z_1\}$ and $\Pi^{(2)} = \{z_2\}$. Then $P_{\mathcal{X}}(\Pi^{(1)}) = P_{\mathcal{X}}(\Pi^{(2)}) = \{x\}$, hence $F(P_{\mathcal{X}}(\Pi^{(1)})) = F(P_{\mathcal{X}}(\Pi^{(2)}))$. Therefore the left-hand sides must be equal: $P_{\mathcal{X}}(N_\phi(\{z_1\})) = P_{\mathcal{X}}(N_\phi(\{z_2\}))$. But $P_{\mathcal{X}}(N_\phi(\{z\}))$ equals $\{x\}$ iff $\phi(z) = 0$ and equals $\emptyset$ iff $\phi(z) = 1$, so $\phi(z_1) = \phi(z_2)$, proving output-determinedness.

($\Leftarrow$) Assume $\phi$ is output-determined. Define an answer-only operator

$$F(A) := \{\, x \in A :\ \tilde{\phi}(x) = 0 \,\},$$

where $\tilde{\phi}$ exists by output-determinedness. For any evidence space $\Pi$, an output assignment $x$ belongs to $P_{\mathcal{X}}(N_\phi(\Pi))$ iff there exists $z \in \Pi$ with $z|_{\mathcal{X}} = x$ and $\phi(z) = 0$, which by $\phi(z) = \tilde{\phi}(x)$ is equivalent to $\tilde{\phi}(x) = 0$. Thus (2) holds. $\qquad\square$

**Remark (instantiated exclusions are typically not output-determined).** In MATLOGIC, exclusions are grounded as "a witness traverses a banned entity" within a scope. Whenever there exist two witnesses $z_1, z_2$ with the same output assignment but different traversal status, $\phi(z_1) \neq \phi(z_2)$, the predicate is not output-determined; hence by Thm. D.1 it cannot be implemented as an answer-only post-filter.

**Proposition D.2** (No scope drift iff out-of-scope branch is safe)**.** *Let $\Pi_1, \Pi_2$ be interface-aligned evidence spaces and let $\phi$ be a predicate. Then*

$$N_\phi\big(D(\Pi_1, \Pi_2)\big) \;=\; D\big(N_\phi(\Pi_1), \Pi_2\big) \tag{3}$$

*if and only if $\forall z \in \Pi_2 :\ \phi(z) = 0$.*

*Proof.* If $\forall z \in \Pi_2, \phi(z) = 0$, then $N_\phi(\Pi_2) = \Pi_2$ and $N_\phi(\Pi_1 \cup \Pi_2) = N_\phi(\Pi_1) \cup N_\phi(\Pi_2) = N_\phi(\Pi_1) \cup \Pi_2$. Conversely, if (3) holds and there exists $z^\star \in \Pi_2$ with $\phi(z^\star) = 1$, then $z^\star \notin N_\phi(\Pi_1 \cup \Pi_2)$ but $z^\star \in N_\phi(\Pi_1) \cup \Pi_2$, contradicting (3). $\quad\square$

**Corollary D.3** (Canonical boundary for EEC exclusions)**.** *Fix an exclusion $d = \langle \texttt{scope\_id}, \phi_d \rangle$ targeting a state $S_t$ with output slots $\mathcal{X}$.*

1. *(**Projection boundary / when**). If $\phi_d$ is not output-determined, then no answer-only operator $F$ satisfies $P_{\mathcal{X}}(N_{\phi_d}(\Pi)) = F(P_{\mathcal{X}}(\Pi))$ for all $\Pi$. Thus exclusion cannot, in general, be postponed past answer projection while preserving semantics.*

2. *(**Disjunction boundary / where**). Consider an interface-aligned disjunctive merge $D(\Pi_1, \Pi_2) = \Pi_1 \cup \Pi_2$ where $\Pi_1$ comes from the alternative containing the targeted state and $\Pi_2$ is out-of-scope. If $\exists z \in \Pi_2$ with $\phi_d(z) = 1$, then $N_{\phi_d}(D(\Pi_1, \Pi_2)) \neq D(N_{\phi_d}(\Pi_1), \Pi_2)$, i.e., applying the exclusion globally after merging is not semantics-preserving.*

*Consequently, any semantics-preserving execution must enforce $\phi_d$ (i) within the alternative/state selected by $\texttt{scope\_id}$, and (ii) before discarding evidence slots needed to evaluate $\phi_d$ (e.g., before projecting to $\mathcal{X}$).*

*Proof.* (1) is Propo. D.1. (2) is Propo. D.2 by violation of the safety condition on $\Pi_2$. $\qquad\square$

# E. Normalization to P→ N→ C/D

Fix an EEC request $\langle \mathcal{T}, E_q, \mathcal{N} \rangle$ with exclusion set $\mathcal{D}$. Executing $\mathcal{T}$ induces contract-visible states $\{S_t\}$ (Def. 2.1). Throughout we prove the normalization under *set semantics* (masking removes candidates; disjunction merges by set union); weighted variants follow by interpreting masking as zeroing weights (support-level equivalence). We view $\mathcal{T}$ abstractly; in our instantiation the executor consumes $\mathcal{T}$ through the normalized control serialization $\text{Serialize}(\mathcal{T}) = \langle \mathcal{P}, \text{SCOPEOF} \rangle$ (App. B). Although Sec. 3.2 represents witness spaces implicitly (endpoint indicators plus provenance), the contract-level composition below is the semantic target that Union/Intersect realize up to existential projection.

At a state $S_t$, let $\Pi_t$ be its evidence space and let $\mathcal{V}_t$ be its slot set. For an executable predicate $\phi$, masking is $N_\phi(\Pi) :=$ $\text{Mask}_\phi(\Pi)$. We write conjunction $\mathbf{C}$ as natural join on shared slots and disjunction $\mathbf{D}$ as union after interface alignment. For each state $S_t$, collect all exclusions targeting it:

$$\mathcal{D}_t := \{\, \langle \texttt{scope\_id}, \phi_d \rangle \in \mathcal{D} \ : \ \texttt{scope\_id}(d) = \texttt{scope\_id}(S_t) \,\}.$$

Define the combined violation predicate

$$\Phi_t(z) := \bigvee_{d \in \mathcal{D}_t} \phi_d(z),$$

so that enforcing all $d \in \mathcal{D}_t$ is equivalent to a single mask $N_{\Phi_t}$.

**Proposition E.1** (Normalization to **P→ N→ C/D**). *Assume (i) each exclusion predicate is state-local (C1), and (ii) execution respects the canonical boundary (App. D): an exclusion is enforced only within its scoped alternative/state and is not postponed past answer projection when it is not output-determined. Then there exists an EEC-compliant execution equivalent to $\mathcal{T}$ in which, for every state $S_t$, all exclusions targeting $S_t$ are enforced* immediately after $\Pi_t$ is produced *and* before $\Pi_t$ is *consumed by any composition operator (**C** or **D**). Equivalently, each in-scope alternative admits the normalized order*

$$\boldsymbol{P} : \Pi_t \text{ produced} \quad \rightarrow \quad \boldsymbol{N} : \Pi_t \leftarrow N_{\Phi_t}(\Pi_t) \quad \rightarrow \quad \boldsymbol{C/D} : \text{compose},$$

*and the final answer set $A_{\text{ans}}$ (and witnesses for trace) is unchanged.*

*Proof.* We apply semantics-preserving rewrites.

**Step 1 (fuse per-state exclusions).** Since masking keeps candidates that violate none of the predicates,

$$\Big( \prod_{d \in \mathcal{D}_t} N_{\phi_d} \Big)(\Pi) \ = \ N_{\Phi_t}(\Pi), \qquad \Phi_t = \bigvee_{d \in \mathcal{D}_t} \phi_d,$$

where the product denotes sequential application in any order (idempotent/commutative under set semantics).

**Step 2 (push masking upstream across conjunction within scope).** Let $\mathbf{C}(\Pi_a, \Pi_b)$ be natural join and let $\phi$ reference only slots present in $\Pi_a$ (state-locality when $\phi$ targets the state that produced $\Pi_a$). Then

$$N_\phi\big(\mathbf{C}(\Pi_a, \Pi_b)\big) = \mathbf{C}\big(N_\phi(\Pi_a), \Pi_b\big),$$

because a joined witness survives iff its $\Pi_a$ component survives and it is compatible with some witness in $\Pi_b$.

**Step 3 (stop before disjunctive merges).** Consider a disjunctive merge $\mathbf{D}(\Pi_1, \Pi_2) = \Pi_1 \cup \Pi_2$, where $\Pi_2$ is out-of-scope for an exclusion targeting the alternative producing $\Pi_1$. By Propo. D.2 (App. D), pushing the mask above the merge is semantics-preserving only if the out-of-scope branch is "safe": $\forall z \in \Pi_2, \phi(z) = 0$. Without this strong condition, the canonical boundary requires the mask to remain on the in-scope branch, i.e., the upstream push stops before $\mathbf{D}$.

**Step 4 (stop before answer projection when not output-determined).** If $\phi$ is not output-determined, Propo. D.1 rules out any answer-only operator that can replace pre-projection masking; hence the mask cannot be postponed past answer projection. Enforcing it earlier at the producing state is valid since the required evidence slots are still available.

**Conclusion.** Starting from any EEC-compliant placement of each exclusion, repeatedly apply Step 2 to push its mask upstream along the in-scope path until it reaches the producing state, and stop before any out-of-scope $\mathbf{D}$-merge (Step 3) or forbidden projection crossing (Step 4). Fuse same-state masks via Step 1. The resulting execution is in the stated **P→ N→ C/D** normal form and preserves $A_{\text{ans}}$ and witnesses. $\square$

*Table 6.* Execution traces for 14 canonical query templates under the P–N–C/D paradigm. Negation is realized as set difference via scope-local masking before any merge/projection (Sec. 2.4, Sec. 3.2).

| Type | Structure | P–N–C/D trace (final set) |
|------|-----------|---------------------------|
| **1p** | $P_1$ | $P_1$ |
| **2p** | $P_1 \to P_2$ | $P_1, P_2$ |
| **3p** | $P_1 \to P_2 \to P_3$ | $P_1, P_2, P_3$ |
| **2i** | $P_1 \land P_2$ | $P_1, P_2, \mathbf{C}\,(X_1 \cap X_2)$ |
| **3i** | $P_1 \land P_2 \land P_3$ | $P_1, P_2, P_3, \mathbf{C}\,(X_1 \cap X_2 \cap X_3)$ |
| **2u** | $P_1 \lor P_2$ | $P_1, P_2, \mathbf{D}\,(X_1 \cup X_2)$ |
| **up** | $(P_1 \lor P_2) \to P_3$ | $P_1, P_2, \mathbf{D},\ P_3$ |
| **ip** | $(P_1 \land P_2) \to P_3$ | $P_1, P_2, \mathbf{C},\ P_3$ |
| **pi** | $P_1 \to (P_2 \land P_3)$ | $P_1, P_2, P_3, \mathbf{C}$ |
| **2in** | $X \land \neg Y$ | $P_X, P_Y, \mathbf{N}\,(X \setminus Y), \mathbf{C}$ |
| **3in** | $X \land Y \land \neg Z$ | $P_X, P_Y, P_Z, \mathbf{N}\,((X \cap Y) \setminus Z), \mathbf{C}$ |
| **inp** | $(X \land \neg Y) \to P$ | $P_X, P_Y, \mathbf{N}\,(X \setminus Y), \mathbf{C},\ P$ |
| **pin** | $P \to (X \land \neg Y)$ | $P, P_X, P_Y, \mathbf{N}\,(X \setminus Y), \mathbf{C}$ |
| **pni** | $P \to (\neg X \land Y)$ | $P, P_X, P_Y, \mathbf{N}\,(Y \setminus X), \mathbf{C}$ |

*Table 7.* Alternative placements of exclusion enforcement (**N**) relative to witness production (**P**) and composition (**C/D**). This table concerns *enforcement placement* (not a literal serial execution order). Conditions from App. D: (OD) output-determined post-filterability; (SAFE) no scope drift across disjunction.

| Order | Applicability / limitation |
|-------|----------------------------|
| **P→N→C/D** | *Always valid* for scoped exclusions: enforce on intermediate evidence within the targeted scope before any merge/projection (Sec. 2.4, App. D–E). |
| **P→C/D→N** | Valid for *negation-free* queries. With negation, semantics is preserved only if (OD) and (SAFE) both hold; otherwise it changes semantics (App. D). |
| **N→P→C/D** | Not meaningful if **N** denotes *enforcement* on produced witnesses. (Mask *compilation* can be hoisted, but mask *application* cannot precede **P**.) |
| **N→C/D→P** | Ill-posed: **C/D** consumes witness evidence produced by **P**. |
| **C/D→N→P** | Ill-posed for the same reason. |
| **C/D→P→N** | Ill-posed; also corresponds to post-hoc filtering after witness consumption. |

**Remark.** Prop. E.1 is implementation-agnostic: under the EEC boundary (App. D), scoped exclusions admit an equivalent execution in the normal form $\mathbf{P} \to \mathbf{N} \to \mathbf{C/D}$. Tables 6–7 instantiate this principle for standard CQA templates by explicitly listing witness-producing steps, the compiled mask step when negation is present, and subsequent composition; deviations from the normal order require additional restrictive assumptions consistent with App. D.

## F. Template Catalog and Operator-Order Sanity Checks

This appendix collects the canonical CQA template library used throughout the paper and provides concrete instantiations of the P→N→C/D normalization. It serves two purposes: (i) define the 14 representative query structures used in experiments (Fig. 4); (ii) provide template-level execution traces under the P–N–C/D paradigm (Table 6) and sanity checks for alternative operator orders (Table 7).

**Fourteen canonical query structures (Fig. 4).** Figure 4 describes the 14 representative logical structures used in our experiments: {1p, 2p, 3p, 2i, 3i, ip, pi, 2u, up, 2in, 3in, inp, pin, pni}. These templates specify (i) the branching topology (conjunction/disjunction) and (ii) the placement of local negation operators, which together determine the required execution boundary for exclusions (Sec. 2.4).

**Concrete P–N–C/D traces per template (Table 6).** Table 6 instantiates the abstract normalization result (App. E) on the above 14 templates by explicitly listing: (i) witness-producing steps $P_1, P_2, \ldots$ (one per atomic producer), (ii) the compiled masking step **N** whenever negation is present, and (iii) subsequent composition by within-group disjunction and across-group conjunction (C/D). This table is meant to be read together with Sec. 3.2: each $P_i$ corresponds to PRODUCEEVIDENCE for one atom, **N** corresponds to scope-local COMPILEMASK/APPLYMASK, and C/D corresponds to UNION/INTERSECT followed by projection.

**Notation for negation rows.** For templates with a negated subquery (e.g., 2in/3in/inp/pin/pni), the post-**N** composition

should be understood as *removing* candidates supported only by the negated branch, i.e., set-difference semantics such as $X \cap \neg Y$ (equivalently $X \setminus Y$), rather than a plain intersection $X \cap Y$. In our executor, this is exactly what the compiled structural mask enforces during **N** before any merge/projection (Sec. 3.2).

**Operator-order sanity checks (Table 7).** Table 7 enumerates alternative operator orders and indicates which query types they can support *only under additional restrictive assumptions* (e.g., global negation or special single-branch negation). This table realizes the non-commutativity boundaries proved in App. D and motivates why P→N→C/D is the only order that works uniformly across all 14 templates without extra assumptions.

Given a query instance, first match it to a canonical structure in Fig. 4. Then read Table 6 to obtain the corresponding P–N–C/D trace and the expected placement of **N**. Finally, Table 7 explains which alternative orders would require stronger assumptions and thus are not generally semantics-preserving under the EEC boundary.

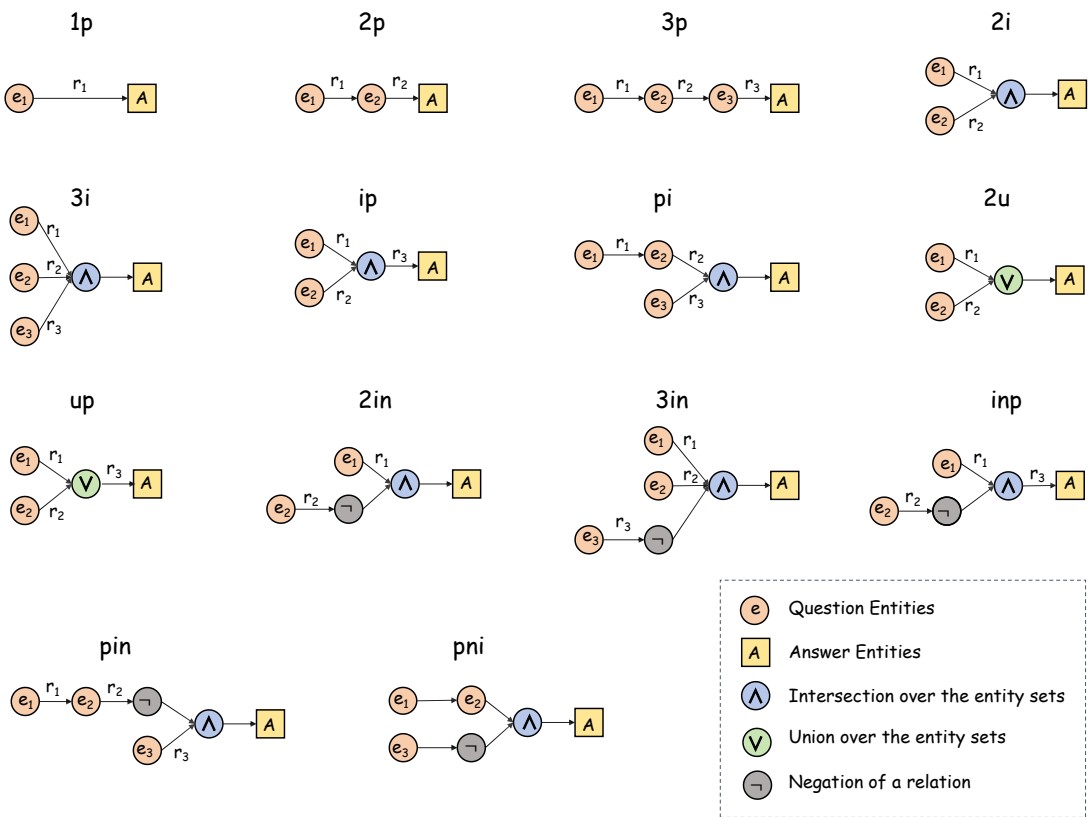

*Figure 4.* Fourteen representative complex query structures used in our experiments.

# G. Workflow Overview of LLM-MATLOGIC

Figure 5 illustrates the end-to-end linear workflow of LLM-MATLOGIC. Starting from a natural-language query, the LLM performs semantic parsing to identify anchor entities and to decide whether scoped negation is required. The system then retrieves a compact multi-node reasoning subgraph around the anchors and grounds the detected negation into an excluded-entity set within this subgraph (consistent with the scoped-negation metadata used in the main text). Given the retrieved subgraph, MatLogic converts it into a sparse-matrix representation and executes a deterministic structural reasoning procedure: it produces multi-hop reachability evidence, applies a structural mask to block witnesses that traverse excluded entities, and finally composes intermediate results via conjunction and disjunction to obtain the answer entities. The figure provides an implementation-oriented view of the interfaces and information flow between the LLM and MatLogic across stages, while the formal contract interface and operator semantics are defined in the main text.

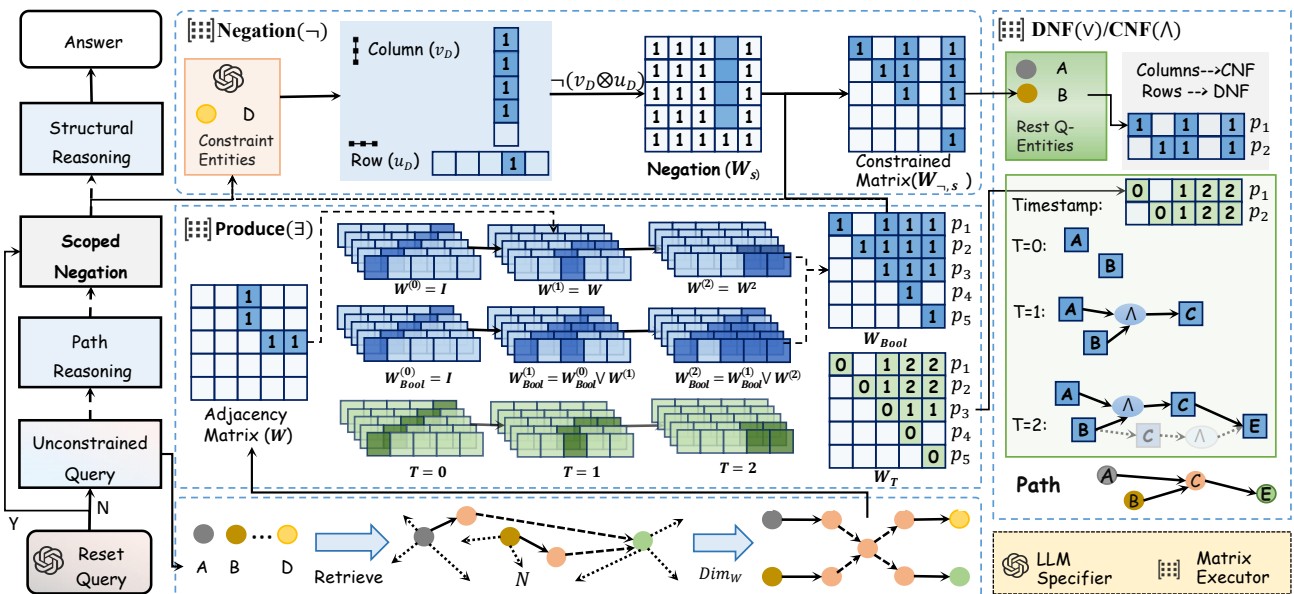

*Figure 5.* **LLM-MATLOGIC workflow.** Matrix-based execution over a retrieved subgraph with scoped negation that excludes the constrained set $D$ via structural masking.

*Table 8.* Statistics of knowledge graph datasets.

| Dataset | #Entities | #Relations | #Triples | Split (train/valid/test) |
|---|---|---|---|---|
| Freebase | $\approx$44M | $\approx$2K | $\approx$2.4B | official RDF dump |
| FB15k-237 | 14,541 | 237 | 310,116 | 272,115 / 17,535 / 20,466 |
| NELL995 | 9,959 | 200 | 114,934 | 82,714 / 11,890 / 20,330 |

## H. Experimental Details

### H.1. Knowledge Graphs

We employ three standard knowledge graphs in our experiments. As shown in Table 8, all three graphs have official train/valid/test edge splits. Query generation and subgraph retrieval are performed within the corresponding edge sets.

- **Freebase (full)** (Bollacker et al., 2008): Originally maintained by Metaweb/Google (2007–2016), Freebase contains approximately 44 million entities and 2.4 billion triples covering people, locations, works, organizations, and more. It is released under Creative Commons in RDF/N-Triples format.

- **FB15k-237** (Toutanova et al., 2015): Derived from Freebase-15k by removing high-frequency and inverse relations to mitigate test leakage. Widely used for KG embedding and query-answering evaluation.

- **NELL995** (Carlson et al., 2010): Extracted by the NELL (Never-Ending Language Learning) system, featuring about 10K entities and 200 relations. Its high sparsity and open-domain nature provide a challenging testbed for generalization.

### H.2. Datasets

**CQA Datasets**. We follow the BetaE recipe to synthesize test queries for 14 CQA patterns: **(1) Template Sampling:** For each pattern, search the training-edge graph for paths matching the structural template. **(2) Filtering & Resampling:** Remove "trivial" queries whose answers all appear in the training set.

**KGQA Datasets.** We conduct experiments on two widely used KBQA benchmarks: **CWQ** (Talmor & Berant, 2018) comprises complex, multi-hop questions over Freebase subgraphs, requiring on average 2–3 relation hops per query. The full

dataset contains 34,689 train / 7,131 valid / 7,334 test questions. In our evaluation, we randomly sample and use *1,000 test questions*. **WebQuestionsSP (WebQSP)** (Yih et al., 2016) extends WebQuestions with SPARQL annotations for Freebase, containing 4,737 train / 1,639 valid / 2,032 test questions. Each example maps to one or more SPARQL queries. In our experiments, we randomly sample and use *1,000 test questions*.

### H.3. Baseline Models

**CQA Task.** We compare three interaction paradigms—*Embedding*, *Prompting*, and *KGQA-driven*—with a total of nine baselines.

- **GQE** (Hamilton et al., 2018) Maps entities and queries into a low-dimensional vector space. Projection is implemented as an affine transform, and intersection as either a dot-product or element-wise min/max.

- **Query2Box** (Ren et al., 2020) Represents query answer sets as axis-aligned hyperrectangles (boxes). Projection is modeled by box translation and scaling; intersection by box overlap; and union via a DNF transformation over multiple box intersections.

- **BetaE** (Ren & Leskovec, 2020) Embeds entities and queries as Beta distributions with parameters $(\alpha, \beta)$. Logical conjunction, disjunction, and negation are implemented via neural operators on these distribution parameters, enabling quantification of uncertainty.

- **HQE** (Choudhary et al., 2021) Embeds entities and queries in the Poincaré ball model of hyperbolic space. Logical operators—projection, intersection, and union—are implemented via hyperbolic affine maps and geometry.

- **HypE** (Choudhary et al., 2021) Uses the hyperboloid model to represent entities, and performs dynamic multi-step query reasoning along hyperbolic geodesics, improving precision on complex paths.

- **CQD** (Arakelyan et al., 2021) Decomposes a complex first-order logic (FOL) query into single-operator subqueries. Each subquery is answered independently by an embedding model (e.g., Query2Box), and the results are fused via a $t$-norm or attention mechanism.

- **CQD-Hybrid** (Gregucci et al., 2024) Decomposes a complex query into a series of atomic subqueries, each processing a single projection or join operation. This approach addresses the tendency of neural link predictors to over-rely on memorized training patterns during inference and thereby enhances the accuracy and robustness of complex query answering while preserving the inherent flexibility of CQD.

- **LARK** (Choudhary & Reddy, 2023) *Complex Logical Reasoning over Knowledge Graphs using Large Language Models.* Retrieves the subgraph containing the query's entities and relations, then constructs a single unified prompt that encodes the full abstract query and its logical operators.

- **MFC** (Zhang et al., 2025) *What is a Good Question? Assessing Question Quality via Meta-Fact Checking.* Decomposes each query into a set of "meta-facts" (KG triples). GPT-3.5-turbo is prompted to verify the truth and quality of each meta-fact, and the candidate answers are then re-ranked based on these verifications.

**KGQA Task.** We compare against four KGQA baselines, including two supervised KBQA systems and two LLM+KG reasoning methods.

- **DiFaR**[2] (Baek et al., 2023) directly retrieves KG facts by embedding questions and facts into a shared representation space, followed by a reranking stage over top retrieved facts.

- **DecAF** (Yu et al., 2023) jointly decodes direct answers and executable logical forms, combining the flexibility of answer generation with the structural grounding of logical-form execution.

- **ToG** (Sun et al., 2024) *Think-on-Graph: Deep and Responsible Reasoning of LLM on Knowledge Graph.* Retrieves a multi-hop subgraph (typically depth 3). At each reasoning step, the current subquery and the evolving graph context are fed into the LLM, which "thinks on the graph" via updated prompts—emphasizing responsible, auditable multi-step inference.

- **MFC (multi-hop application)** (Zhang et al., 2025) Applies the full MFC framework to multi-hop QA tasks: meta-facts are extracted from the retrieved subgraph, verified by the LLM, and re-ranked to produce the final answers.

```
RESET_QUERY_PROMPT = """
You are a knowledge-graph query parsing agent.

Task
Given the Question, extract:
1) [ENTITIES]: entity IDs explicitly mentioned in the question (entity IDs only, not relation
IDs).
2) [NEGATION]: whether the question contains any exclusion/negation constraint.
3) [NEGATED_CONSTRAINT]: the shortest verbatim span(s) in the question that express the
exclusion/negation.

Output contract (strict)
- Output EXACTLY one <RESULT>...</RESULT> block and nothing else.
- Use the exact field names shown below.
- IDs must be integers; deduplicate; sort ascending.
- If no entity IDs are found, output [].
- [NEGATION] must be exactly Y or N.
- If [NEGATION] = N, then [NEGATED_CONSTRAINT] must be None.
- If multiple exclusion spans exist, join them with "; " and keep each span verbatim (do NOT
paraphrase).

<Question>
{question}
</Question>

<RESULT>
[ENTITIES] [<id>, <id>, ...]
[NEGATION] <Y|N>
[NEGATED_CONSTRAINT] <verbatim span(s) or None>
</RESULT>
"""
```

```
SCOPED_NEGATION_PROMPT = """
You are a knowledge-graph pruning agent.

Goal
From the given Constraint text and the provided candidate set (path entities), return
which candidate entity IDs must be removed.

Inputs
- Constraint: {constraint}
- Path entities (candidate set): {path}

Rules (strict)
- Work ONLY with the candidate set; never invent IDs outside it.
- Remove an ID only if it is explicitly mentioned in the Constraint under a
negative/exclusion expression.
- Accept ID mentions formatted as "entity <ID>" or bare "<ID>".
- Output a JSON list of integers; deduplicate; sort ascending.
- If no explicit IDs are mentioned in the Constraint, output [].

Output contract (strict)
- Output EXACTLY one <RESULT>...</RESULT> block and nothing else.

<RESULT>
[REMOVE_ENTITIES] [<id>, <id>, ...]
</RESULT>
"""
```

*Figure 6.* The prompt template in LLM-MATLOGIC.

## H.4. Subgraph Extraction

To keep the context within a controllable size, we do not feed the entire knowledge graph (KG) to the model. Instead, we extract a local subgraph by performing query-driven $k$-hop neighborhood expansion based on the entities and relations appearing in the query.

Let $\tau$ denote a query type, and let $E_\tau^1$ and $R_\tau^1$ be the initial sets of entities and relations that occur in the query, respectively. The first-layer neighborhood $N_1$ is constructed by collecting KG triples that satisfy the constraints induced by $E_\tau^1$ and $R_\tau^1$. For layer $k$ ($k \geq 2$), we update the entity and relation sets using the heads/tails and relations observed in the previous neighborhood, yielding $E_\tau^k$ and $R_\tau^k$, and then collect the next neighborhood triples to form $N_k$. This recursive expansion yields a $k$-hop subgraph centered around query-relevant entities/relations, which is then used as structured context for prompting and reasoning.

To control the subgraph size and the input length, we impose two constraints: (1) the expansion depth $k$ is determined by the query type (e.g., for 3-hop query types we set $k = 3$); and (2) we stop the expansion once the accumulated subgraph context exceeds the token budget of the underlying LLM, ensuring the context remains admissible.

## H.5. Structured Ranking

In the CQA setting, MatLogic's matrix execution typically produces an *unordered* set of reachable candidate answers. However, Mean Reciprocal Rank (MRR) requires an *ordered* list. To validate the reliability of matrix execution (MatLogic), we avoid using language-model probability outputs and instead compute candidate scores purely from graph-structural evidence. We therefore adopt a fixed, non-learned structured ranking scheme that maps the reachable leaf-node candidates to a ranked list and then reports MRR.

Under the $A_{\text{ans}}$ mode, we first obtain the set of reachable leaf-node candidate entities. For each candidate entity $e \in A_{\text{ans}}$, we compute two structural quantities based on the query's reasoning graph:

**Path support (path_count$(e)$).** This is the number of *distinct reasoning paths* from the query root to $e$, reflecting the strength of independent logical support. Intuitively, a candidate visited by more independent reasoning paths has stronger structural evidence.

**Path depth (hop$(e)$).** This is the *shortest* logical distance (minimum number of hops) from the query root to $e$. Intuitively, candidates reachable via shorter reasoning chains have more direct structural support.

We define the final structured ranking score as:

$$S(e) = \frac{\text{path\_count}(e)}{\text{hop}(e) + 1}.$$

The "+1" avoids division by zero when hop$(e) = 0$ and applies a smooth penalty to deeper paths. Candidates are ranked in *descending* order of $S(e)$ to form the final ranked list.

For each query, after sorting candidates by decreasing $S(e)$, let $r$ be the rank position (starting from 1) of the *highest-ranked correct answer*. The reciprocal rank is $1/r$, and MRR is computed by averaging over all queries:

$$\text{MRR} = \frac{1}{|\mathcal{Q}|} \sum_{q \in \mathcal{Q}} \frac{1}{r_q}.$$

### H.6. Prompt Template

**RESET\_QUERY\_PROMPT.** We use a query-parsing prompt to convert a natural-language question into structured signals: it extracts entity IDs explicitly mentioned in the question and determines whether any exclusion/negation constraint is present. When negation exists, the prompt also returns the shortest verbatim span(s) expressing the constraint, enabling precise alignment of the natural-language exclusion with the subsequent graph-level operators. The output is restricted to a single <RESULT> block: entity IDs are emitted as a deduplicated, ascending-sorted integer list; the negation flag is Y/N; and the constraint field is fixed to None when no negation is detected.

**SCOPED\_NEGATION\_PROMPT.** We use a pruning prompt to instantiate a negation constraint as a set of entities to remove. The input consists of the constraint text and a candidate set of path entities; the prompt is required to identify only those IDs in the candidate set that are explicitly targeted by a negation/exclusion expression, and it must not output any ID outside the candidate set. The output is restricted to a single <RESULT> block where [REMOVE_ENTITIES] is a deduplicated, ascending-sorted integer list; if the constraint text contains no explicit ID mention, the prompt returns an empty list.

