# OpenReview forum: "LLM-MatLogic: Executable Exchange Contracts for Knowledge-Graph Query Answering with Scoped Negation"
_ICML.cc/2026/Conference — ICML 2026 regular_

### Official Review · Reviewer_j3Kq · 2026-03-12

**Soundness:** 2
**Presentation:** 3
**Significance:** 3
**Originality:** 3
**Overall Recommendation:** 4
**Confidence:** 3

**Summary:**

**One-line Summary**

This paper proposes EEC (Executable Exchange Contract) and its executor, MATLOGIC, for the KGQA task to handle scoped negation correctly.

&nbsp;

**Core Idea**

The authors' core ideas are as follows:

- A front-end specifier passes a "contract" that explicitly defines (1) where the exclusion attaches (scope) and (2) what to exclude (banned set or predicate).
- A back-end executor then implements this as a scope-local mask.
- The execution should be in the order of "Produce → Negate → Compose/Distribute", showing why otherwise can be erroneous.

**Compliance With Llm Reviewing Policy:**

Affirmed.

**Final Justification:**

The rebutttal addressed my main concerns.

**Key Questions For Authors:**

On negation-aware patterns such as inp and pin, LARK remains stronger. Do the authors have a hypothesis for why the proposed framework is less effective on these mixed compositions?

**Limitations:**

yes

**Strengths And Weaknesses:**

**Strengths.**

*(1) Clear problem formulation*

The paper breaks down an important failure mode in LLM-to-KG reasoning,  identifying that natural-language negation is often both scope-sensitive and evidence-dependent. This is a clear and practically relevant problem setting.

&nbsp;

*(2) Novel Interface-level framing*

Rather than treating negation as only a prompting issue or only an executor issue, the paper formulates it as an explicit contract between the specifier and the executor. This method gives the method a clean and reasonable conceptual structure.

&nbsp;

*(3) Good diagnostic evaluation*

The paper separates unknown vs. oracle constraints and varies the specifier while holding the executor fixed, which helps distinguish specification errors from execution errors.

&nbsp;

**Weaknesses.**

*(1) The supported form of negation appears narrower than the paper’s broader framing*

The paper motivates scoped negation in general, including evidence conditions and support-level exclusions. However, LLM-MATLOGIC (Sec. 3.1) appears to be largely centered on scope-indexed banned entity sets B. Since natural-language negation can involve richer predicate-level, relation-level, or attribute-level semantics beyond simple entity exclusion, the current formulation seems narrower than the broader motivation presented in the paper. It is encouraged for the authors to clarify the precise scope of the framework, including which types are not yet handled.

&nbsp;

*(2) It is unclear which component is responsible for the observed gains*

While the paper compares baseline methods under the same retrieved subgraph—which is a reasonable attempt —this may not fully guarantee a fair comparison, since each method uses the graph in a fundamentally different way. Therefore, a more fine-grained performance breakdown seems to be needed to more convincingly explain why the proposed framework is more effective.  Current results only show the advantage of the full pipeline, and it remains unclear which component—e.g., the specifier, the executor, or the overall strategy—is primarily responsible for the observed gains.

&nbsp;

*(3) Limited baseline coverage in NL-KGQA*

While the paper compares against a broader set of embedding- and prompting-based baselines in CQA, the NL-KGQA evaluation is limited to only two baselines (ToG and MFC). This makes it ambiguous to assess how broadly competitive the proposed method is.

---

> ### Author Rebuttal · Authors · 2026-03-30
>
> Thank you for these important comments, and we have provided detailed explanations as follow.
>
> # Q1:LLM-Matlogic vs LARK
>
> In the CQA experiments, **the comparison is not entirely under the fair setting**. The main difference between LLM-MATLOGIC and LARK is that LARK assumes the query structure is already known and directly encodes the full structured query into the reasoning prompt. In contrast, LLM-MATLOGIC first parses a natural-language description rewritten from the query template, and under the **unknown-constraints setting**, it must autonomously compile the negation scope from **mixed query content before performing structured logical computation**. (We will explain this in the revised version.)
>
> To make a fairer comparison, we further introduce a *known constraints* setting, **when the model is explicitly informed that negation exists in the query, LLM-MATLOGIC outperforms LARK**. The comparison is shown below:
>
> | Method | 2in | 3in | inp | pin | pni |
> |---|---:|---:|---:|---:|---:|
> | LARK | 7.5 | 4.8 | 22.3 | 18.7 | 3.6 |
> | Unknown-constraints | 11.4 | 12.7 | 15.1 | 7.5 | 10.9 |
> | Known-constraints | **27.2** | **28.5** | **31.7** | **19.5** | **23.8** |
>
>
>
> # Weakness1
>
> Describing LLM-MATLOGIC as merely performing simple entity deletion is inconsistent with the paper's own formulation. The paper explicitly states that "*B is a grounded instantiation of the exclusion set D*" and that the executor will "*compile each B into a scope-local structural mask.*" Thus, predicate-, relation-, or attribute-level negation is first interpreted by the specifier and grounded into local executable constraints, **which are instantiated in the current implementation as scope-specific banned entity sets**; these are then converted by the executor into structural negation over intermediate witnesses and paths, rather than simple answer-level deletion.
>
>
> # Weakness2:
>
> We further conduct a quantitative analysis on negation-aware query structures.
> The follow table compares three settings: end-to-end LLM-MATLOGIC, MATLOGIC without constraints (providing correct entity recognition and negative entity recognition), and LLM-only on the same retrieved subgraph without structured execution.
>
> | Setting | 2in | 3in | inp | pin | pni |
> |---|---:|---:|---:|---:|---:|
> | LLM-MATLOGIC | 11.4 | 12.7 | 15.1 | 7.5 | 10.9 |
> | MATLOGIC-only | 47.6 | 48.3 | 52.9 | 45.2 | 49.8 |
> | LLM-only (same retrieved subgraph) | 2.7 | 1.5 | 3.2 | 2.2 | 1.7 |
>
> The results show that **LLM-only cannot effectively perform subgraph logical computation with negation when no executor is used**. In contrast, **MATLOGIC-only  outperforms LLM-MATLOGIC when the correct constraints are provided**, indicating that the Matlogic can handle this type of computation reliably. Therefore, **whether EEC can correctly specify the negation constraints has a more direct impact on the final performance than the executor itself**.
>
> To further analyze the impact of negation constraints on the final results, the table below summarizes the **specification-related error symptoms** among failed cases
>
>
> |  | 2in | 3in | inp | pin | pni |
> |---|---:|---:|---:|---:|---:|
> | Negation not identified | 32.8% | 23.7% | 33.1% | 28.5% | 43.2% |
> | Incorrect negated entity | 48.3% | 45.8% | 44.8% | 38.6% | 53.5% |
> | Incorrect exclusion set | 63.1% | 58.6% | 61.4% | 54.2% | 68.4% |
> | Incorrect anchor entity | 12.2% | 18.5% | 12.3% | 21.1% | 18.7% |
>
> These categories are not mutually exclusive. Most errors are concentrated in negated-entity identification and exclusion-set construction, suggesting that negation signals have a important impact on overall accuracy.
>
>
> # Weakness3:
> We further include *RoG* and *Retrieve-Rewrite-Answer* as additional NL-KGQA baselines, with the results shown below.
>
> | Method | CWQ | WebQSP |
> |---|---:|---:|
> | Retrieve-Rewrite-Answer | 52.1 | 71.4 |
> | RoG | 57.9 | 75.2 |
> | LLM-MatLogic | **63.5** | **76.7** |
>
> Moreover, we would like to clarify that the paper does not evaluate natural-language capability through NL-KGQA alone. In the CQA, EEC does not operate directly on explicit symbolic templates. instead, we first rewrite each query structure into a natural-language description, and then let the LLM parse the NL-query content and negation information in a zero-shot prompt.
>
> The CQA experiments provide controlled analysis of EEC across different query structures, while the NL-KGQA experiments further validate the method in more natural-language-oriented scenarios.

---

> > ### Author Rebuttal · Reviewer_j3Kq · 2026-04-04
> >
> > I thank the authors for their detailed rebuttal. My concerns have been well addressed through the additional clarifications and I am pleased to raise my score.

---

### Official Review · Reviewer_kJFr · 2026-03-12

**Soundness:** 2
**Presentation:** 2
**Significance:** 3
**Originality:** 3
**Overall Recommendation:** 3
**Confidence:** 4

**Summary:**

LLM-MatLogic proposes a system for knowledge graph query answering that handles scoped negation through Executable Exchange Contracts (EEC) and a MatLogic executor. The system addresses the challenge that natural-language negation is scope-sensitive (constraining only specific subgoals) and evidence-dependent (targeting only certain supporting paths), which current LLM-to-KG systems frequently mishandle. The EEC encodes exclusions as executable control metadata, while MatLogic compiles these into scope-local masks applied during multi-hop propagation under a unified PN-CID schedule. The work is evaluated on structured complex query answering (CQA) and end-to-end natural-language KGQA, with contract-aligned diagnostics designed to isolate specification errors from execution errors. The authors claim the system ensures exclusions are enforced before witness loss and branch entanglement, providing more accurate handling of negation than existing approaches.

**Compliance With Llm Reviewing Policy:**

Affirmed.

**Key Questions For Authors:**

Why were recent neural-symbolic methods (LMPNN, NSMP, GNN-QE) not included as baselines?
Can you provide quantitative analysis using contract-aligned diagnostics: What percentage of errors come from specification vs. execution? What are the most common failure modes and their frequencies? When does scope-local masking help vs. harm? How does performance vary by query complexity? This analysis is essential for validating the diagnostic methodology's value.

**Limitations:**

yes

**Strengths And Weaknesses:**

Strengths

The paper identifies a genuine and underexplored challenge in production KGQA systems: natural-language negation is scope-sensitive (constraining only specific subgoals/branches) and evidence-dependent (targeting only certain supporting paths), which current systems frequently mishandle. Concrete examples like "movies featuring an actor but not as a cameo" effectively illustrate practical scenarios where scope-sensitivity matters, making the abstract problem tangible and demonstrating clear real-world relevance.

Weaknesses

The paper does not provide rigorous formal definitions for core technical components. The EEC specification format is described informally without formal grammar or type system.

There are no formal proofs or rigorous arguments that the PN-CID schedule correctly enforces scoped negation semantics, that scope-local masking preserves logical equivalence, or that the system is sound with respect to any formal semantics.

Key terms are used extensively throughout the paper but never precisely defined: "witness loss" (What exactly is a witness? When is it lost? What are the formal conditions?), "branch entanglement" (What does this mean formally? How is it detected or prevented?), "scope-local" (How are scope boundaries defined? What makes masking "local" vs. "global"?), and "evidence-dependent" (How does this differ from standard FOL negation semantics?). This imprecision pervades the technical presentation and undermines rigor.

---

> ### Author Rebuttal · Authors · 2026-03-30
>
> Thank the reviewers for these important comments, and we have provided detailed explanations and supplements as follow.
>
> # Q1: New baselines
>
> We have added three more recent neuro-symbolic baselines, namely LMPNN, GNN-QE, and NSMP, and the results are shown in the table. The inclusion of these baselines does not change the main conclusion of the paper: LLM-MatLogic still achieves better performance on most CQA structures, with particularly clear advantages on compositional queries and some negation-aware queries.
> | Method | 1p | 2p | 3p | 2i | 3i | pi | ip | 2u | up |
> |---|---:|---:|---:|---:|---:|---:|---:|---:|---:|
> | LMPNN | 45.9 | 13.1 | 10.3 | 34.8 | 48.9 | 22.7 | 17.6 | 13.5 | 10.3 |
> | GNN-QE | 42.8 | 14.7 | 11.8 | 38.3 | **54.1** | 31.1 | 18.9 | 16.2 | 13.4 |
> | NSMP | 46.7 | 15.1 | 12.3 | 38.7 | 52.2 | 31.2 | 23.3 | 17.2 | 11.9 |
> | LLM-MatLogic | **69.2** | **58.2** | **42.5** | **57.1** | 43.2 | **31.8** | **24.2** | **39.6** | **26.3** |
>
>
> | Method | 2in | 3in | inp | pin | pni |
> |---|---:|---:|---:|---:|---:|
> | LMPNN | 8.7 | 12.9 | 7.7 | 4.6 | 5.2 |
> | GNN-QE | 10.0 | 16.8 | 9.3 | 7.2 | - |
> | NSMP | **11.9** | **17.6** | **10.8** | 7.9 | _ |
> | LLM-MatLogic | 11.4 | 12.7 | **15.1** | 7.5 | **10.9** |
>
>
>
> # Q2: Quantitative analysis
>
> We conducted further quantitative analysis on query structures containing negation and divided the errors into two parts: specification-related errors and executor-side diagnostics.
> (1) **Specification-related errors**
> The table below summarizes the EEC-related errors in the **failed samples**. These categories are not mutually exclusive.
>
> |  | 2in | 3in | inp | pin | pni |
> |---|---:|---:|---:|---:|---:|
> | Negation not identified | 32.8% | 23.7% | 33.1% | 28.5% | 43.2% |
> | Incorrect negated entity | 48.3% | 45.8% | 44.8% | 38.6% | 53.5% |
> | Incorrect exclusion set | 63.1% | 58.6% | 61.4% | 54.2% | 68.4% |
> | Incorrect anchor entity | 12.2% | 18.5% | 12.3% | 21.1% | 18.7% |
>
> (2) **Executor-side diagnostics**
> The table below shows the results of executor-only settings after **providing correct entity recognition and negative entity recognition**, reflecting the execution performance under more accurate input constraints.
>
> | Setting | 2in | 3in | inp | pin | pni |
> |---|---:|---:|---:|---:|---:|
> | LLM-Matlogic | 11.4 | 12.7 | 15.1 | 7.5 | 10.9 |
> | Executor-only | 47.6 | 48.3 | 52.9 | 45.2 | 49.8 |
>
> Some of the current errors do not stem from the executor's execution capabilities, but rather from deviations in the LLM's extraction of construction of exclusion sets.
>
>
> Based on the above analysis, ***negation signals have a important impact on model accuracy**.
>
> In the paper, Table 5 compares model performance under unknown constraints and known constraints, i.e., whether negation is inferred by the LLM or explicitly provided. From unknown constraints to known constraints, the average MRR increases from 0.199 to 0.369, and the average Hits@1 increases from 0.181 to 0.300. This indicates that once the scoped constraints are correctly specified, **the executor can perform the downstream computation reliably**.
>
> # Q3: Scope-local masking
> Sec. 2.4 and App. D-E show not only why scoped negation should be executed locally, **but also when changing the operator order becomes harmful**. Specifically, the normalized P-N-C/D schedule helps by preserving evidence-dependent negation and preventing scope drift across branches. In contrast, if negation is moved before projection, the intermediate evidence required to evaluate the exclusion is no longer available; if it is moved after disjunctive merge, a scope-local negation may be incorrectly turned into a global one. Therefore, when negation depends on intermediate evidence or constrains only a specific branch, a local mask must be applied within the corresponding scope and before merge/projection.
>
>
> # Q4: Performance vary by query complexity
>
> The model performance is mainly affected by two factors: first, the experimental results show that performance declines as query **depth or width** increase. Secondly, queries with negation are also generally more challenging than standard path, intersection, or union queries. Furthermore, tab 4 and 5 show that the **LLM's semantic parsing accuracy and reasoning ability** also directly affect the final results.
>
>
> # Weakness1&2
>
> These two weaknesses are not accurate. In Sec. 2, the paper already provides explicit formal definitions of EEC and its intermediate states, and further specifies executability, scope binding, and verifiability through C1-C3(Sec. 2.3). Moreover, Sec. 2.4 and App. D-E formally justify that scoped negation must be enforced within the corresponding scope, via the non-commutativity theorems for projection and Conjunction/Disjunction together with the normalization result.

---

### Official Review · Reviewer_TNuq · 2026-03-13

**Soundness:** 3
**Presentation:** 2
**Significance:** 3
**Originality:** 2
**Overall Recommendation:** 4
**Confidence:** 2

**Summary:**

The paper introduces LLM-MATLOGIC, a framework for knowledge-graph question answering (KGQA) that explicitly models scoped negation in translating natural language queries. To address the limitations of existing LLM-based KGQA pipelines that often mishandle negation semantics, the authors propose Executable Exchange Contract (EEC) interface to separate query specification from deterministic execution. The executor, MATLOGIC, performs matrix-based reasoning over the knowledge graph while enforcing scope-aware negation constraints before projection and composition operations. Experiments show that the proposed method improves accuracy on negation-involving queries compared with several embedding-based and neural-symbolic baselines, while also requiring fewer LLM calls during inference.

**Compliance With Llm Reviewing Policy:**

Affirmed.

**Key Questions For Authors:**

See the weaknesses.

**Limitations:**

Yes

**Strengths And Weaknesses:**

Strengths:
1. The LLM-MatLogic applies a clean conceptual separation to let the LLM construct the contract, and deterministic executor to perform the operation. This helps isolates semantic parsing errors and improve interpretability.
2. The paper provides a formal theoretical analysis of negation placement and show that negation cannot be postpone after projection. This analysis helps justify the proposed execution order and clarifies an often overlooked issue in LLM-based KGQA systems.
3. The proposed method can have good computational efficiency. MatLogic itself runs in sub-second time on the subgraphs and matrix-based propagation is simple and scalable.

Weakness:
1. While the paper proposes the EEC abstraction and a scoped-negation execution strategy, the architecture closely follows the reasoning-on-graph paradigm adopted in prior work [1], where an LLM first converts a natural-language query into a structured representation and a deterministic engine executes the reasoning over the knowledge graph. For example, frameworks such as RoG [1] similarly separate semantic parsing from graph execution and observe that most errors arise from the LLM-generated reasoning specification rather than the executor. In this context, the primary novelty of the current work appears to lie in the explicit handling of scoped negation and the ordering of execution operations, rather than introducing a fundamentally new reasoning framework. As a result, the conceptual contribution may be viewed as incremental relative to existing neural-symbolic KGQA pipelines.
2. The synthetic CQA dataset uses template-generated queries, which may not fully capture the complexity and ambiguity of negation phenomena in real natural-language questions. Additional evaluation on more realistic or naturally occurring queries would strengthen the empirical validation.
3. The writing can be improved for clarity in several sections. The paper introduces heavy formalism and dense definitions early on, which makes some parts difficult to follow. The core ideas and intuition behind the framework could likely be communicated more clearly with additional explanations or examples

[1] Reasoning on graphs: Faithful and interpretable large language model reasoning. arXiv preprint arXiv:2310.01061.

---

> ### Author Rebuttal · Authors · 2026-03-30
>
> We thank the reviewer for these helpful comments and appreciate the opportunity to clarify these points.
>
> # Q1: EEC vs RoG
> The difference between EEC and RoG  not only in the surface workflow, but also in two aspects: the core problems they solve and their execution paradigms.
> **Core problem**: RoG mainly emphasizes planning answer paths through multi-round iterative reasoning, whereas EECfocuses on structured logical computation over the retrieved subgraph, especially the modeling and execution of **negation constraints**.
> **Execution paradigm**: EEC performs structured logical computation directly on the retrieved subgraph, without repeated iteration, and requires fewer LLM calls than RoG.
>
> We used CWQ to perform NL-KGQA testing, and the results are as follows:
>
> | Method | Hits@1 | Calls |
> |---|---:|---:|
> | RoG | 57.9 | 12.4 |
> | LLM-MatLogic | 63.5 | 3.0 |
>
>
> # Q2:  Template-generated queries
>
> In the CQA experiments, **EEC is not built on traditional explicit query templates, and its logical computation does not rely on predefined query structures**. Instead, we rewrite each query structure into a natural-language description, and then let the LLM identify the negation information and query content in a zero-shot prompt. Therefor, **EEC operates on natural language rather than directly exposed symbolic templates**.
>
> Under this setting, the 14 CQA query structures are mainly used for controlled analysis of EEC's effectiveness across different structures, while the KGQA experiments further validate the feasibility of EEC in more natural-language scenarios.
>
> We will clarify this setup more explicitly in the revised version.
>
>
> # Q3: Writing
> In the revised version, we will further improve the writing to enhance the overall readability.

---

> > ### Author Rebuttal · Reviewer_TNuq · 2026-04-04
> >
> > Thank you to the authors for the clarifications.
> >
> > 1. The rebuttal helps clarify the distinction in terms of problem focus (negation handling vs. planning) and execution paradigm (single-pass structured execution vs. iterative reasoning). I agree that the proposed method provides a more structured and efficient execution mechanism, particularly for handling negation constraints. However, my original concern was primarily about the overall architectural paradigm, where both approaches follow a similar LLM->structured-query->executor pipeline. While the differences are meaningful at the level of execution design, I still view the contribution as an extension of existing reasoning-on-graph frameworks rather than a fundamentally new paradigm.
> > 2. I appreciate the clarification that query structures are rewritten into natural-language descriptions. However, the rebuttal does not provide details on the diversity of these natural-language formulations (e.g., different phrasings of negation such as “not,” “except,” “without,” or more implicit forms). Since the proposed framework relies on the LLM to correctly identify scoped negation and map it into structured constraints, its performance may be sensitive to how negation is expressed linguistically. It would be helpful to include examples of the natural-language templates used, as well as an analysis of robustness to paraphrasing or variation in negation expressions. Without such evaluation, it remains unclear how well the method generalizes to real-world queries with diverse linguistic forms.
> > 3. I appreciate the authors’ willingness to improve the presentation, which would make the paper more accessible.
> >
> > The rebuttal provides helpful clarifications but does not fundamentally change my evaluation. I still find the paper technically sound with a meaningful contribution in scoped negation handling, though somewhat incremental in terms of overall framework novelty. I will maintain my original score.

---

> > > ### Author Response · Authors · 2026-04-07
> > >
> > > # Q1:
> > > Thank you for the further clarification. We agree that, at the macro architectural level, our method still belongs to the broad **LLM - structured query - executor** family. What we would like to emphasize is that the key difference from prior work lies not mainly in the high-level pipeline, but in the **semantic-logical decoupling paradigm** adopted by our framework.
> > >
> > >
> > > Specifically, we explicitly separate **LLM-based natural-language semantic parsing** from **structured logical computation over the retrieved subgraph**: the former is responsible for identifying query semantics and negation constraints, while the latter performs deterministic logical reasoning. Importantly, our framework explicitly formalizes complex logical computation over the retrieved subgraph into a unified **P-N-C/D** execution paradigm, which is further validated in our CQA experiments. To the best of our knowledge, such an explicit formalization of the structured execution process for **scoped negation** has been **less directly articulated in prior work**.
> > >
> > > Therefore, our contribution is not merely an implementation-level refinement within the existing broad pipeline, but rather makes the **representation, scope binding, enforcement boundary, and composition order of scoped negation** explicit as an **interface-level execution paradigm**.
> > >
> > >
> > >
> > > # Q2:
> > >
> > > Thank you for raising this important point. In our setup, the template-based CQA questions are rewritten into natural-language descriptions rather than exposing symbolic negation directly. For negation statements, we use the more implicit phrase "other than" in our current templates (relevant templates in the source code), rather than the explicit **"not"** or **"without"**. For example, our current rewrites mainly use formulations *"entities connected to \(e_2\) by any relation other than \(r_2\)"*. The analysis (200 samples) of different negation words is as follows:
> > >
> > > | Setting | 2in | 3in | inp | pin | pni | Avg |
> > > |---|---:|---:|---:|---:|---:|---:|
> > > | Not (MRR) | 0.466 | 0.272 | 0.375 | 0.286 | 0.392 | 0.358 |
> > > | Negation identified | 100% | 99.5% | 100% | 100% | 100% | 99.9% |
> > > | without (MRR) | 0.415 | 0.241 | 0.346 | 0.279 | 0.324 | 0.321 |
> > > | Negation identified | 99% | 94.5% | 98% | 97.5% | 96.5% | 97.1% |
> > > | other than (MRR) | 0.162 | 0.187 | 0.238 | 0.213 | 0.197 | 0.199 |
> > > | Negation identified | 87.5% | 91.5% | 93% | 92.5% | 91.5% | 91.2% |
> > >
> > > The results suggest that, under the zero-shot setting, **"other than"** is more difficult to recognize than **"not"** or **"without"**, as it conveys negation in a more implicit way. At the same time, the framework is already robust to **more explicit negation expressions**, indicating that the structured execution itself can work reliably when the negation cue is clearly expressed.
> > >
> > > In our CQA setting, our negation is mainly based on relatively fixed **"other than"** templates to test a less explicit negation cue. Therefore, we primarily use **CQA** as a **controlled setting** to test whether scoped negation can be correctly attached and enforced under different logical structures. To provide additional evidence in more natural-language environments, we also conduct end-to-end **NL-KGQA** experiments on **CWQ** and **WebQSP**.
> > >
> > > At the same time, the gap between **known** and **unknown** constraints (tab 5) suggests that, within the EEC framework, performance improves when negation is identified more accurately, which also points to an important direction for future work.
> > >
> > >
> > > # Q3:
> > > Thank you for the helpful suggestion. This will help us further improve the clarity and presentation of the paper.

---

### Official Review · Reviewer_ziP9 · 2026-03-15

**Soundness:** 3
**Presentation:** 2
**Significance:** 4
**Originality:** 3
**Overall Recommendation:** 4
**Confidence:** 3

**Summary:**

The paper introduces a framework for handling scoped negation in knowledge-graph query answering (KGQA) using large language models (LLMs). It addresses the limitations where natural-language negation is scope-sensitive and evidence-dependent, often leading to incorrect exclusions. The core contribution is the Executable Exchange Contract (EEC) that explicitly specifies scope-bound exclusions as executable metadata exchanged between an LLM specifier (which compiles natural-language queries into structured requests) and a matrix-based executor called MATLOGIC. The system is evaluated on structured complex query answering (CQA) benchmarks (FB15k-237 and NELL995) and end-to-end natural-language KGQA (CWQ and WebQSP), demonstrating superior performance on negation-aware patterns (e.g., MRR improvements of 3-8 points over baselines like LARK and BetaE) and compositional queries, while introducing diagnostics to isolate specifier vs. executor errors.

**Compliance With Llm Reviewing Policy:**

Affirmed.

**Key Questions For Authors:**

The paper mainly focuses on the set logic, which is acceptable in most cases. But I would like to hear more about how the EEC applies to cyclic constraints that we cannot decompose into set projection/masking/disjunction/conjunction directly.

**Limitations:**

no. see my key questions for authors.

**Strengths And Weaknesses:**

Strength:
- The EEC provides a clear separation between query specification and execution, making negation handling explicit, scope-aware, and verifiable, which reduces ambiguity in LLM-based pipelines.
- The empirical performance is somewhat good.
- Introduces contract-aligned diagnostics to attribute failures to the specifier or executor, offering actionable insights beyond end-to-end metrics.

Weakness:
- The method is mainly developed on the answer set notation, and the negation is defined. It lacks some discussion about the cyclic queries.
- Some important baselines are missing, as the method focuses on scoped negation. Table 2 is particularly important, and only limited baselines are listed. Some baselines that are worth mentioning in particular, negation:
- Some presentations could be improved. For example, in the introductory example in section 2.1, there is no definition of $\phi$, but directly used in both the equation and the main content.


For their discussions on cyclic queries and the performance in negation queries with embeddings, I would also want to mention two previous works.
- https://arxiv.org/pdf/2301.08859
- https://arxiv.org/pdf/2304.07063

---

> ### Author Rebuttal · Authors · 2026-03-30
>
> We sincerely thank the reviewer for the careful reading and valuable suggestions. Our responses to the raised questions are provided below.
>
>
> # Q1: Answer set notation and Cyclic queries.
>
>
> **Answer set notation**: We clarify that EEC is not designed to operate directly on final answer sets. Instead, its logical computation is performed in the intermediate states of the retrieved subgraph structure. In particular, answer negation involves posterior exclusion of the final answer set, while structural negation involves constraints on intermediate evidence or subgraph structure, determining which supporting paths remain valid during the reasoning process.
>
> **Cyclic queries**: Natural language query templates are typically tree-like or DAGs-like queries, rather than explicitly cyclic queries. The 14 query structures in our experiments also follow this standard setting, primarily to verify the effectiveness and reliability of our structured logic computation method under standard CQA formulas.Furthermore, EEC does not execute projection, intersection, union, or negation directly on symbolic query templates. Instead, **it first parses a natural-language query description, locates relevant entities, and then performs unified structured logical computation over the retrieved subgraph**. We will further clarify this scope in the revised version.
>
>
>
> # Q2:Important baselines focus on scoped negation.
>
> We will use LMPNN and FIT as additional baseline methods. Both of these methods assume that the query template is known, while LLM-MatLogic parses the natural language query and then performs structured logic computation on the subgraph. Therefore, the problem settings are not entirely the same. To make a fairer comparison, we further introduce two additional settings: unknown constraints and known constraints. Experimental results on FB15K-237 are as follows:
>
> | Method | 2in | 3in | inp | pin | pni |
> |---|---:|---:|---:|---:|---:|
> | FIT | 14.0 | 20.0 | 10.2 | 9.5 | - |
> | LMPNN | 8.7 | 12.9 | 7.7 | 4.6 | 5.2 |
> | Unknown-constraints | 11.4 | 12.7 | 15.1 | 7.5 | 10.9 |
> | Known-constraints | **27.2** | **28.5** | **31.7** | **19.5** | **23.8** |
>
> The results indicate that LLM-MatLogic performs best when the negation constraints are explicitly provided (Known-constraints). More importantly, under the more realistic Unknown-constraints setting, it still outperforms LMPNN overall and exceeds FIT on a subset of query types.

---

> > ### Author Rebuttal · Reviewer_ziP9 · 2026-04-04
> >
> > My questions are answered, and I am glad to see that the limitations in cyclic queries are properly discussed and will appear in the future.

---

### Decision · Program_Chairs · 2026-04-30

**Decision:**

Accept (regular)

**Comment:**

Reviewers agree that the paper addresses a real KGQA failure mode and that the proposed scoped-negation execution strategy using LLMs is a significant contribution. The rebuttal strengthened the manuscript by adding stronger NeSy and NL-KGQA baselines, providing diagnostics that separate specification from execution errors, and clarifying that performance is largely bottlenecked by the identification of negation constraints. The main remaining concerns are that the overall pipeline is still incremental relative to prior work, evaluation is still somewhat controlled/narrow, and robustness to varied natural-language negation remains only partially validated. Given the reviewers' recommendation, this seems borderline but slightly above the bar for acceptance.